# Pupil-linked phasic arousal evoked by violation but not emergence of regularity within rapid sound sequences

Sijia Zhao [1], Maria Chait [1], Fred Dick [2,3], Peter Dayan [4], Shigeto Furukawa[5] & Hsin-I Liao[5]

The ability to track the statistics of our surroundings is a key computational challenge. A prominent theory proposes that the brain monitors for unexpected uncertainty – events which deviate substantially from model predictions, indicating model failure. Norepinephrine is thought to play a key role in this process by serving as an interrupt signal, initiating model-resetting. However, evidence is from paradigms where participants actively monitored stimulus statistics. To determine whether Norepinephrine routinely reports the statistical structure of our surroundings, even when not behaviourally relevant, we used rapid tone-pip sequences that contained salient pattern-changes associated with abrupt structural violations vs. emergence of regular structure. Phasic pupil dilations (PDR) were monitored to assess Norepinephrine. We reveal a remarkable specificity: When not behaviourally relevant, only abrupt structural violations evoke a PDR. The results demonstrate that Norepinephrine tracks unexpected uncertainty on rapid time scales relevant to sensory signals.

[1] Ear Institute, University College London, London WC1X 8EE, UK. [2] Department of Psychological Sciences, Birkbeck College, London WC1E 7HX, UK. [3] Department of Experimental Psychology, University College London, London WC1H 0DS, UK. [4] Max Planck Institute for Biological Cybernetics, 72076 Tübingen, Germany. [5] NTT Communication Science Laboratories, NTT Corporation, Atsugi 243-0198, Japan. Correspondence and requests for materials should be addressed to M.C. (email: m.chait@ucl.ac.uk)

A growing body of work demonstrates that observers maintain detailed models of the statistics of their environments over various timescales, combining this information with sensory input to inform choice[1], increase response accuracy[2], speed up reaction times[3,4], and improve detection[5,6]. A key challenge in this context is keeping track of the evolving input statistics so as to ensure model validity. Here we investigated automatic and controlled aspects of the neural response to this challenge in a fast-paced domain.

For effective model maintenance, a central dilemma faced by the brain is to arbitrate between gradual and abrupt changes in the environment[7]. In the former case, modelupdating progresses at a steady pace, dictated by the model's estimate of local noise (expected uncertainty) arising from tracked environmental stochasticities[8,9]. However, environments can also change substantially and suddenly. The ability to detect such change points is crucial for optimal behavior, because they indicate that the observer's beliefs about the environment are no longer a valid representation of reality, and should be reset[4,10]. For example, in Nassar et al.[11] subjects were instructed to predict sequentially presented numbers drawn from a Gaussian distribution whose mean occasionally changed abruptly. Following change points, participants tended to alter their behavior in a way that reflected abandonment of old expectations and more rapid acquisition of new ones - e.g., recent events had more influence on decisions than those occurring further in the past, equivalent to an increased learning (and forgetting) rate.

Although such change processes can often be described optimally in hierarchical probabilistic terms, an alternative heuristic is for the brain to monitor events that fall outside the threshold of expected uncertainty estimated for the model, and treat them as signaling potential change points in the environment. Such so-called 'unexpected uncertainty'[12] has been suggested as interrupting top-down processes so as to prioritize bottom-up evidence accumulation, thereby speeding up discovery of the new structure of the environment[9,12,13]. The neuromodulator norepinephrine (alternatively noradrenaline or NE) has been hypothesized to play a critical role in this updating process[4,10,12,14,15]. NE is generated in the brainstem nucleus Locus Coeruleus (LC), which projects extensively across the brain and spinal cord[13,16,17] and is thus optimally placed to signal a global state change in the environment. However, a vast literature has also implicated NE in controlling vigilance, orienting behavior, selective attention, and surprise[13,18,19], suggesting that it might instead play a much less specific role, associated with regulating arousal.

The bulk of work on NE and model updating in humans has involved paradigms in which participants actively monitor the statistics of the stimulus, for instance through explicit tracking tasks[2,10,11,20] or in speeded stimulus-response paradigms[4]. It is therefore an open question whether NE involvement is driven by behavioral relevance[21] (for decisions or motor responses), or if NE plays a more ubiquitous role in reporting changes in the statistical structure of our surroundings. In the latter case, we need to examine which events trigger its release.

Sensory systems continuously analyze probabilistic information which unfolds on a rapid timescale, even when this information is not immediately relevant to behavior[5,22,23]. It is therefore compelling to ask (1) how the fast-paced and automatic mechanisms that detect changes in statistics within rapid sensory signals interface with NE, (2) how NE's involvement compares with other aspects of neural dynamics, and (3) what effect there is, if any, of making the changes behaviorally consequential. In addition, by understanding the contingencies to which NE responds, we hope to gain extra clarity on the heuristic separation between gradual and abrupt change that is critical for effective model maintenance.

To examine these questions, we sought a sensory paradigm that induces such changes, along with a way of assessing the effect on NE and other neural systems. For the first, we considered rapid auditory patterns (Fig. 1a) consisting of sequences of tone-pips (new on each trial) containing transitions either from a repeating, regular (REG) to a random (RAND) frequency structure, or the reverse. These stimuli are particularly suited for our purposes since at a presentation rate of 20 Hz, the sequences are too rapid for naive listeners to explicitly follow the unfolding pattern. Rather, the changes in sequence structure (in both directions) readily pop out from the stimulus stream irrespective of subjective effort (see stimulus examples in Supplementary Audio 1–4). Furthermore, the changes induce patterns of neural dynamics[22,24] which hint that they might illuminate the central dilemma about suddent versus gradual change. Transitions from regular to random frequency structures evoke a mismatch neural response, triggered by the abrupt violation of the regular pattern. The opposite transitions, random-to regular – despite having matched overall spectro-temporal structure and being similarly detectable – do not generate a mismatch response. Instead, the dynamics of the brain response are consistent with an evidence accumulation process which changes more slowly from one structure to the other.

In order to assess NE, we turned to the eyes. Indirect measures of NE release can be obtained from monitoring non-luminance-mediated changes in pupil size[18,25]. This renders pupillometry an attractive, non-invasive means of probing NE activity in the brain. There is a consistent mechanistic correlation between spiking activity in the LC and changes in pupil size, both when spontaneously occurring, and when triggered by external events[25,26]. In particular, transient pupil dilation responses (PDR) have been shown to causally relate to phasic activity within the LC-NE system[25,26], though there remains uncertainty about the specific circuitry[27]. Capitalizing on these links, recent pupillometry studies have revealed a relationship between pupil dilation and predictability[2,11], providing (indirect) evidence for the involvement of phasic LC-NE responses in signaling uncertainty.

Thus, we monitored pupil size whilst subjects listened to changing auditory sequences, including the disambiguating regular-random (REG-RAND) and random-regular (RAND-REG) transition types. If the pupil-linked LC-NE system generally monitors for salient state changes in the environment, both transition types are expected to evoke PDRs. However, under the hypothesis that phasic LC-NE responses are selective for abrupt changes even when detected automatically in speeded inputs, we should observe pupil dilation responses to the former, but not the latter, transition. Consistent with the hypothesized role of the LC-NE system in perceptual model updating, we provide converging results from multiple experiments which demonstrate that the pupil-linked LC-NE system automatically and selectively tracks unexpected uncertainty on rapid timescales, relevant to sensory processing, and outside of behavioral relevance.

## Results

**Exp1: PDR to violation but not emergence of regularity.** The basic stimulus set is shown in Fig. 1a. In referring to the stimuli we adopt a nomenclature where the term in uppercase denotes the type of signal (RAND vs REG) and the subscript indicates to the size of the sub-pool from which the relevant pattern is created. Thus, $RAND_{20}$ is a tone series created by randomly selecting each tone (with replacement) from a full pool of 20 frequencies. $RAND_{10}$ is a series created from a subset of 10 different frequencies (randomly selected from the full pool), while $REG_{10}$ is a regular pattern consisting of a repeating sequence of 10 tones (a different pattern, and a different sub-pool, on each trial).

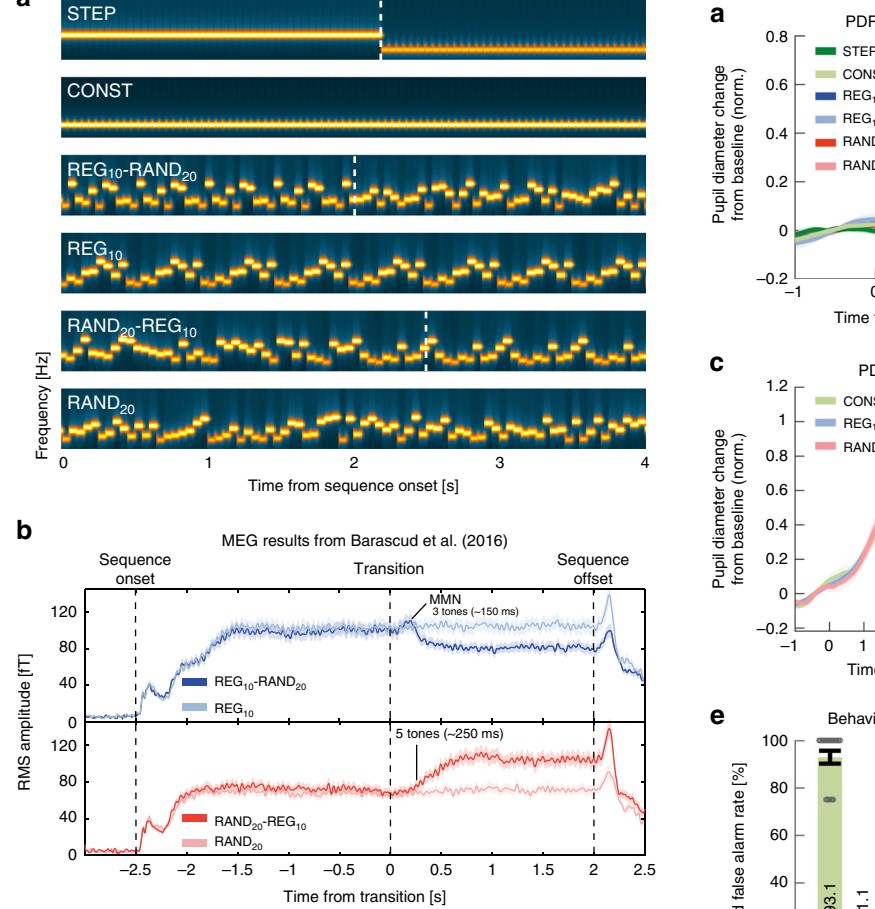

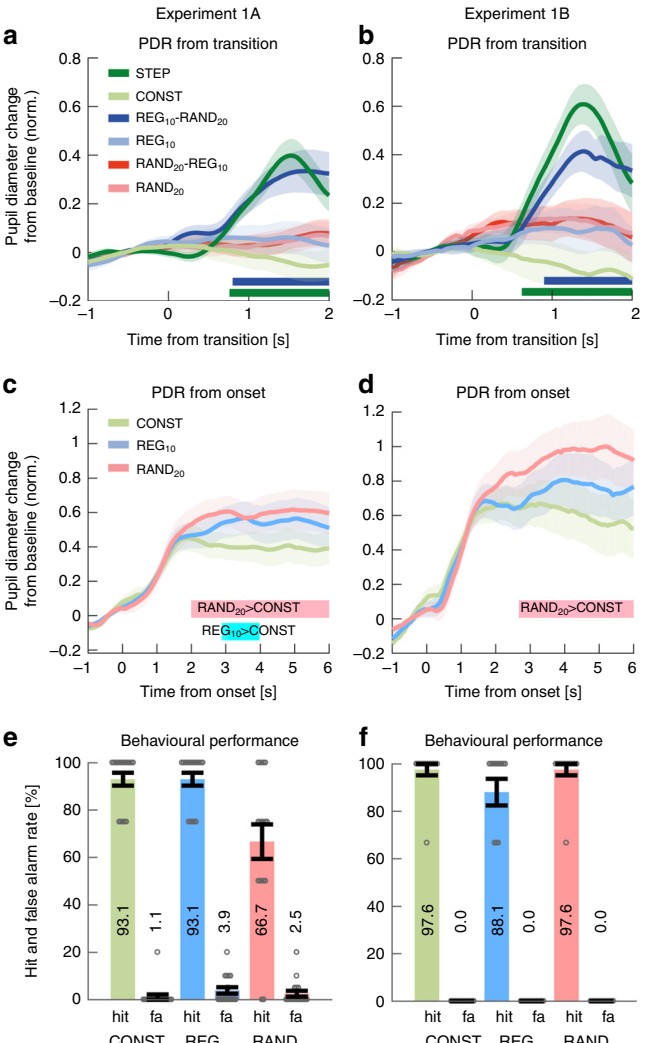

**Fig. 1** Basic stimuli and brain responses recorded with MEG. **a** The stimuli were sequences of concatenated tone-pips (50 ms) with frequencies drawn from a pool of 20 fixed values. The tone-pips were arranged according to six frequency patterns, generated anew for each subject and on each trial: CONST sequences consisted of a single repeating tone; STEP contained a step change from one tone frequency to another; $REG_{10}$ sequences were generated by randomly selecting 10 frequencies from the pool and iterating that sequence to create a regularly repeating pattern; $RAND_{20}$ were generated by randomly sampling from the full pool with replacement; $REG_{10}$-$RAND_{20}$ and $RAND_{20}$-$REG_{10}$ sequences contained a transition between a regular and random pattern or vice versa. Transition times (between 2.5 and 3.5 s post onset) are indicated by a white dashed line. In $RAND_{20}$-$REG_{10}$ sequences, the transition time is defined as occurring after the first full regularity cycle, i.e. once the transition becomes statistically detectable. For presentation purposes only, the plotted sequence lengths are equal. Durations varied randomly between 5 and 7 s. **b** Brain responses ($n = 13$) to $REG_{10}$-$RAND_{20}$ (top panel) and $RAND_{20}$-$REG_{10}$ (bottom panel), together with their no-change controls, recorded with magnetoencephalography (MEG). Plotted is Root Mean Square (RMS) over channels, as an estimate of instantaneous power. The figures show the entire stimulus epoch, relative to the transition. Shaded areas are ±1 SEM. The transition from RAND to REG is associated with a gradual increase in sustained power from ~250 ms (5 tones) post transition. The transition from REG to RAND evokes an MMN-like response (at ~150 ms after the transition) followed by a sharp drop in the sustained response. These changes in power are hypothesized to reflect the instantiation (RAND-REG) or interruption (REG-RAND) of a contextual top-down model. See Barascud et al.[22] for more details. See also Supplementary Figure 1 for an information theoretic[28] characterization of the stimuli: the REG-RAND transition evokes an immediate increase in information content (IC) of post-transition tones. In contrast, because any REG pattern is equally likely under RAND, a gradual decrease in IC is seen for RAND-REG transitions as the model 'discovers' the predictable structure within the sequence

Brain response (MEG and EEG) data suggest that while $REG_{10}$-$RAND_{20}$ and $RAND_{20}$-$REG_{10}$ transitions are characterized by opposite statistics (emergence vs. violation of regularity; see also Supplementary Fig. 1 for an information theoretic characterization of the sequences[28]), both are detected automatically, and at a similar latency even when participants' attention is directed elsewhere[22,24]. When asked to respond behaviorally to transitions, listeners exhibit ceiling performance and similar reaction times with comparable variability[22] (also replicated here in Exp3A). Thus, these signals are well suited for dissociating transitions associated with model resetting from those associated with model updating and provide an elegant method for disambiguating the role of the pupil-linked LC-NE system in tracking statistics of rapidly evolving sensory signals.

To control for overall engagement[29] and ensure broad attention to the auditory stimuli, but without requiring active tracking of the transitions, naïve participants (in Exps 1, 2, and 4) detected short silent gaps as they listened to the tone-pip sequences. Gap occurrence was uncorrelated to state transition, and the subset of sequences containing gaps were excluded from analyses. (In Exp3 we investigate the effect of making the transition task-relevant).

Exp1A ($n = 18$): Fig. 2a plots the average pupil size data across all participants as a function of time relative to the transition. Clear PDRs were observed in the STEP and $REG_{10}$-$RAND_{20}$ conditions, but not in the $RAND_{20}$-$REG_{10}$ condition.

**Fig. 2** REG-RAND but not RAND-REG transitions are associated with a PDR. **a** Average pupil diameter over time relative to the transition in Experiment 1A ($n = 18$). Solid lines represent the average normalized pupil diameter. The shaded area shows ±1 SEM. Color-coded horizontal lines at graph bottom indicate time intervals where cluster-level statistics show significant differences between each change condition and its no-change control. In STEP, the pupil diameter started to increase around 300 ms post-transition, reaching peak amplitude at 1520 ms; it statistically diverged from its control, CONST, from 760 ms through to sequence offset. Similarly, the PDR to $REG_{10}$-$RAND_{20}$ increased from ~700 ms post-transition, peaking at 1720ms. $REG_{10}$-$RAND_{20}$ statistically diverged from its control, $REG_{10}$ from 800 ms post-transition through to sequence offset. No significant differences between $RAND_{20}$-$REG_{10}$ and $RAND_{20}$ were observed. **b** Average pupil diameter over time relative to the transition in Experiment 1B ($n = 14$, replicating Experiment 1A). The divergence of STEP from its control, CONST, was significant from 620 ms post-transition. $REG_{10}$-$RAND_{20}$ significantly diverged from its control, $REG_{10}$, from 900 ms. As in Experiment 1A, no significant differences were observed between $RAND_{20}$-$REG_{10}$ and RAND20 throughout the epoch. **c, d** Average pupil diameter over time, relative to sequence onset. Colored lines indicate time intervals where cluster-level statistics showed significant differences between conditions. There were no significant differences between $RAND_{20}$ and $REG_{10}$ in either experiment. In Experiment 1B $REG_{10}$ showed a significantly larger pupil diameter than CONST between 2880 and 3960 ms post-onset. In both experiments, $RAND_{20}$ was associated with a significantly larger pupil diameter than CONST from 2000ms (Expt1A) and from 2680 ms (Expt1B) post-onset. Importantly there were no differences between $RAND_{20}$ and $REG_{10}$ in either experiment. **e** Behavioral hit and false alarm (FA) rates for the gap detection task in Experiment 1A. Gray circles represent individual participant data, and error bars are ±1 SEM. Performance on $RAND_{20}$ was significantly reduced relative to the other conditions. **f** Gap detection task in Experiment 1B. Lengthening the gap in Experiment 1B resulted in equated performance across conditions

Performance on the gap detection task was good overall (Fig. 2e) but we observed a main effect of condition on hit rates (arc-sine transformed for these and all subsequent statistical analyses on hit rates, $F(1.198,20.372) = 8.285$, $p = 0.007$). Post hoc tests confirmed that hit rate in $RAND_{20}$ was lower than in CONST ($p = 0.026$) and $REG_{10}$ ($p = 0.026$), while CONST and $REG_{10}$ did not differ significantly ($p = 1.0$).

To assure that performance disparities were not driving differential PDR effects, gap duration in Exp1B ($n = 14$) was lengthened by 50 ms to equate task performance across conditions. The revised paradigm successfully eliminated performance differences between conditions (Fig. 2f), with a repeated-measures ANOVA showing no effect of stimulus condition on hits ($F(2,26) = 2.115$, $p = 0.141$) or false positive rates ($F(2,26) = 1.0$, $p = 1.0$). The PDR pattern observed in Exp1A was entirely replicated (Fig. 2b). Overall, the results of Exp1 confirmed that PDRs are consistently evoked by STEP and $REG_{10}$-$RAND_{20}$ transitions, but not $RAND_{20}$-$REG_{10}$ transitions.

Figure 2c, d present average pupil diameter in the no-transition stimuli from sequence onset as measured in Exp1A and 1B. No significant differences were observed between $RAND_{20}$ and $REG_{10}$ in either Exp1A or 1B, suggesting similar average pupil diameter before the transition. Identical results (no pre-transition difference between $RAND_{20}$ and $REG_{10}$) were also obtained in Exp2 and 4 below. Overall, the results suggest the absence of substantial pre-transition effects. The regression approach used to remove the variance related to the pre-transition pupil size (see Methods) also assures that the observed post-transition PDR effects are not driven by baseline differences between conditions.

**The null effect is not due to temporal spread.** To confirm that the null effect for $RAND_{20}$-$REG_{10}$ indicates the absence of a pupil response and is not instead a consequence of an increased temporal spread of dilation events, pupil dilation (PD) and constriction (PC) rates were also analyzed (see Methods). This analysis is fundamentally different from the PDR analysis in that it focuses on the incidence of PD (or PC) events, irrespective of their amplitude, and therefore provides a sensitive measure of subtle changes in pupil dynamics potentially evoked by the transitions. Figure 3 shows pupil dilation events from each trial, for each subject ($n = 32$, combining Exp1A & B), over an interval of 2 s before to 2 s after the transition.

STEP and $REG_{10}$-$RAND_{20}$ transitions were associated with an increase in PD rate shortly after the transition, whereas no such change in rate was observed for $RAND_{20}$-$REG_{10}$. This effect was also mirrored in the constriction data, confirming that neither PD nor PC dynamics changed following $RAND_{20}$-$REG_{10}$ transitions. Results were equivalent across event-duration thresholds of 75 and 300 ms (see Methods). Overall, this set of analyses provides further evidence for a null PDR response related to the $RAND_{20}$-$REG_{10}$ transition.

**Exp2: pupil responses to 'pure' pattern violations.** To interpret this first set of results, it is important to establish whether the PDR observed for STEP and $REG_{10}$-$RAND_{20}$ transitions revealed 'true' sensitivity to pattern violations, or rather was driven by low-level stimulus changes (frequency deviants). In Exp1, at least half of $REG_{10}$-$RAND_{20}$ trials involved the appearance of a novel frequency at the time of transition. This is also trivially the case for all STEP trials. It is therefore possible that the PDR reflects a simple response to the detection of a new frequency in the stimulus. In Exp2 the stimulus set (Fig. 4) was amended to include conditions where the transition was manifested as a change in pattern with or without frequency deviants.

Figure 5 plots all the conditions which contained a regular-to-random transition: all evoked a marked PDR relative to the $REG_{10}$ control. Notably a prominent PDR was observed for $REG_{10}$-$RAND_{10}$, i.e. a transition from a REG to a RAND pattern manifested as a change in pattern only, while maintaining the same 10 frequencies. In contrast, no significant difference was observed for any of the random-to-regular transitions (Fig. 5d). This was also the case for the $RAND_{10}$-$REG_{10d}$ condition where the RAND and REG sequences differed in frequency content in addition to the change in pattern. Whilst a small peak is visible in that condition, no significant differences are observed when compared to the no-transition $RAND_{10}$ condition (Fig. 5c).

To confirm that the various REG and RAND conditions did not diverge pre-transition, we analyzed the pupil response from stimulus onset (Fig. 5e) and found no significant differences between any of the conditions. Consistent with Exp1, behavioral performance did not differ across conditions (Fig. 5f).

**Exp3: behavioral relevance induces a PDR to RAND-REG.** Exp1 and Exp2 measured responses to transitions when they were not behaviorally relevant. To understand the effect of task relevance on PDRs, we introduced an active behavioral transition-tracking task. In Exp3A ($n = 14$) the experimental conditions were as in Exp1, but participants were asked to detect pattern changes rather than silent gaps.

Behavioral results are summarized in Fig. 6a. Hit rate data demonstrated that all transition conditions were highly detectable by human listeners. Although false positive rates were all low, there was a main effect of condition ($F(1.405,18.262) = 15.272$, $p < 0.001$), where, consistent with previous work[22], there was a small but significantly higher false positive rate for (no-transition)

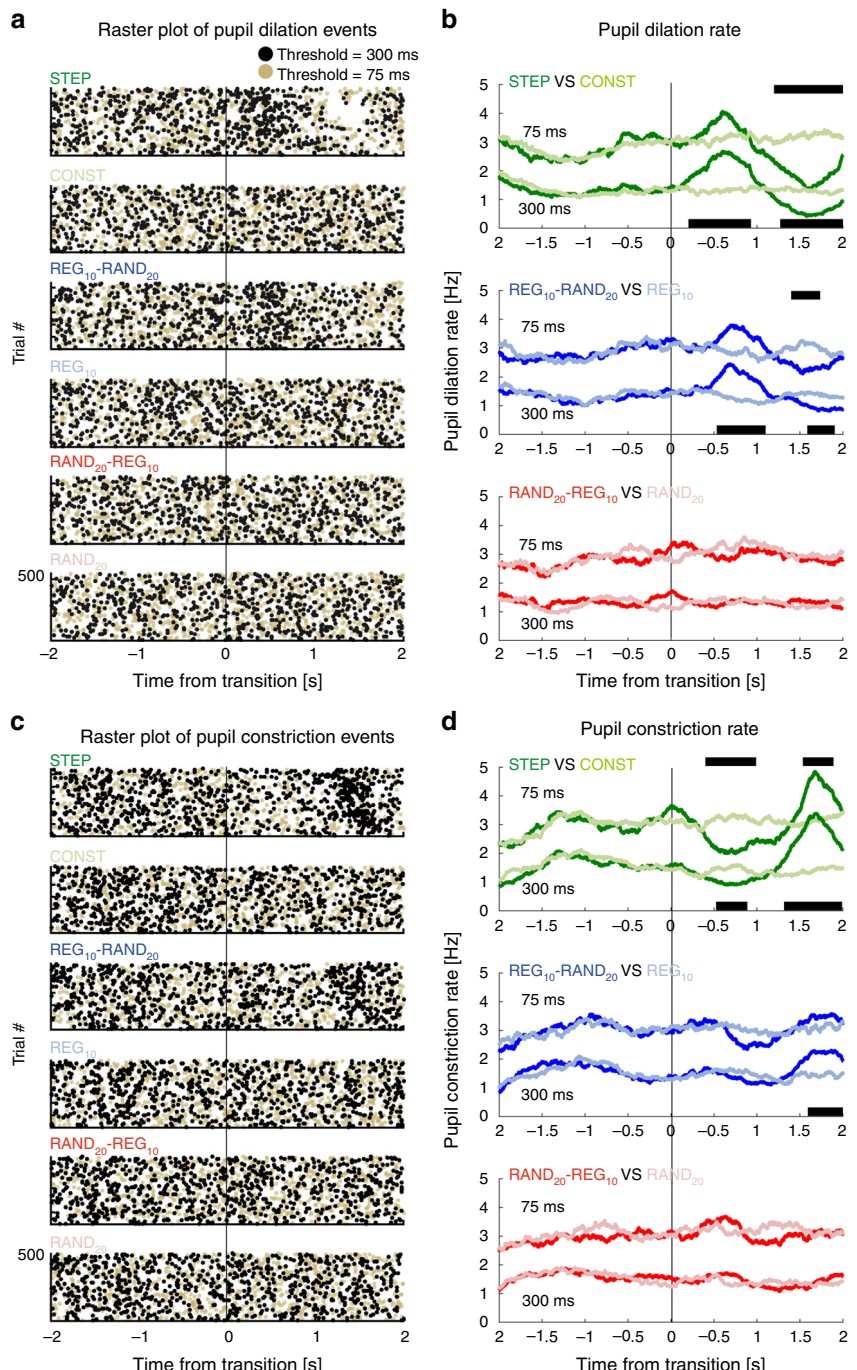

**Fig. 3** Experiment 1: pupil dilation and constriction rates. **a** Raster plots of pupil dilation (PD) events extracted from all trials and all participants (collapsed over Experiment 1A & B). Each line represents a single trial. Black dots represent the onset of a pupil dilation with a duration of at least 300 ms, yellow dots represent pupil dilation onsets with a threshold duration of 75 ms. Transition time is indicated by a black vertical line. **b** Pupil dilation rate (running average with a 500-ms window) as a function of time relative to the transition. The black horizontal lines indicate time intervals where cluster-level statistics showed significant differences between each change condition and its no-change control. The statistics for the PD rates with a threshold duration of 75 ms and 300 ms are placed above and below the graph, respectively. **c**, **d** present the pupil constriction (PC) rate results, arranged in the same format

$RAND_{20}$ compared with CONST and $REG_{10}$, with the latter two not differing significantly ($p = 0.083$; Bonferroni corrected). To avoid confounds due to false positive disparities, all false positive trials were excluded from pupil analysis.

For reaction time, a repeated measures ANOVA confirmed a main effect of condition ($F(2,26) = 90.723$, $p < 0.001$; STEP < REG-RAND < RAND-REG), consistent with previous work[22].

Turning to Pupillometry, we observed three differences relative to Exp1:

(1) $RAND_{20}$-$REG_{10}$ also evoked a PDR: Fig. 6b plots the average pupil diameter relative to the transition. Clear PDRs were observed in all three change conditions. Critically – and unlike the previous experiments with a gap detection task – a robust PDR was associated with $RAND_{20}$-$REG_{10}$ when the transition was task-relevant.

(2) Delayed PDR peak for $REG_{10}$-$RAND_{20}$: compared to Exp1 and 2, we also observed a substantial shift in the latency of the PDR to $REG_{10}$-$RAND_{20}$ (Fig. 6c). Active transition

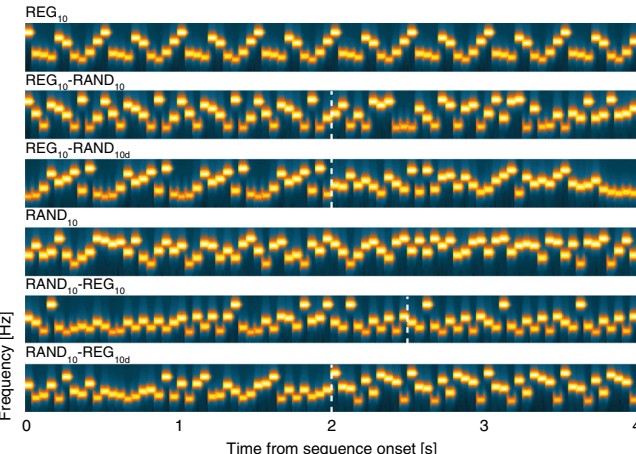

**Fig. 4** Example spectrograms for the additional stimuli used in Experiment 2. $REG_{10}$ and $RAND_{10}$ were generated by randomly selecting 10 frequencies from the pool and then either iterating that sequence to create a regularly repeating pattern or presenting them in random order. $REG_{10}$-$RAND_{10}$ and $RAND_{10}$-$REG_{10}$ sequences were created from the same 10 frequencies (different sets on each trial). Thus, the transition was manifested as a change in pattern only, without the occurrence of a frequency deviant. $REG_{10}$-$RAND_{10d}$ and $RAND_{10}$-$REG_{10d}$ were created such that the frequencies used for the REG and RAND portions of the sequence were different (non-overlapping sets of 10 frequencies each). The transition was thus manifested as both a change in pattern, and also as a change in frequency content. The stimulus set also included $REG_{10}$-$RAND_{20}$, $RAND_{20}$-$REG_{10}$ and $RAND_{20}$ sequences (identical to those in Experiment 1). Dashed vertical white lines indicate the transition time. Note that for $RAND_{10}$-$REG_{10}$ the transition time is defined as occurring after the first full regularity cycle (see also Fig. 1). The transition time is not adjusted for $RAND_{10}$-$REG_{10d}$ because the transition becomes statistically detectable immediately when the alphabet changes (at the nominal transition time). For presentation purposes, the plotted sequence lengths are equal, but experimental sequences durations varied randomly between 6.0 and 7.5 s

detection slowed the PDR by ~300 ms, with a peak latency of 1840ms in Exp3 relative to 1400–1600ms in the previous experiments. Peak latency to STEP did not change.

(3) Differences in pupil diameter observed from sequence onset: In Exp1 and Exp2 (see also Exp4 below) we consistently observed no difference between $REG_{10}$ and $RAND_{20}$ when analyzing pupil responses relative to sound onset. In contrast, in Exp3 we observed a pre-transition disparity between $REG_{10}$, $RAND_{20}$ and CONST (Fig. 6d), such that $RAND_{20}$ sequences evoked the largest sustained pupil dilation, followed by $REG_{10}$. The sustained pupil diameter for $RAND_{20}$ was significantly higher than $REG_{10}$ from 3080 ms post-onset. This cannot be explained by the higher false alarm rate of $RAND_{20}$, as trials with incorrect responses were excluded from analysis, but may be a consequence of the computational demands or perceptual effort associated with tracking $RAND_{20}$ sequences (see Discussion).

We explored the relationship between reaction time (RT) and pupil diameter across single trials (Fig. 7). Generally, peak pupil dilation (indicated by hot colors in Fig. 7a) occurred about 1s after the button press[30]. This relationship is also evident in Fig. 6e: the onset of the PDR to each of the transitions closely coincides with button press timing.

The fact that peak PDR occurs substantially after the button press may indicate that the behavioral response itself either triggers or otherwise modulates the PDR. To understand the relationship between RT and the PDR, we examined whether button press timing was systematically linked to key measures of PDR dynamics. We tested the relationship between RT and the maximum PDR amplitude (Fig. 7b) on a single-trial basis (see Methods). In the STEP condition, there was no significant association between RT and maximum PDR amplitude ($t(84.86) = 1.29$, $p = 0.1996$); this also held true in the $REG_{10}$-$RAND_{20}$ condition ($t(234.6) = 0.22$, $p = 0.8232$). However, in the $RAND_{20}$-$REG_{10}$ condition, RT was significantly positively associated with maximum PDR amplitude ($t(334.9) = 3.06$, $p = 0.0024$): here, RT was estimated to account for 2.8% of variance in the maximum PDR amplitude (estimate of partial R2 derived using an implementation of the Nakagawa and Schielzeth[31] algorithm, see Methods).

We then asked whether RT was associated with the timing of the maximum PDR (Fig. 7c). In the STEP condition, there was a small but significant association between RT and PDR latency ($t(297) = 1.97$, $p = 0.0496$, accounting for an estimated 1.5% of PDR latency variance. (Note, however, this association was not significant when slopes were allowed to vary). In the $REG_{10}$-$RAND_{20}$ condition, RT was also significantly associated with PDR latency ($t(308.8) = 4.64$, $p < 0.0001$), and accounted for an estimated 7.2% of its variance. Finally, RT was also associated with PDR latency in the $RAND_{20}$-$REG_{10}$ condition ($t(310.8) = 4.85$, $p < 0.0001$), and an estimated 6.3% of RT variance.

Finally, we asked if RT was associated with the timing of the maximum derivative of the PDR (i.e. the time at which the rate of change of pupil size is maximal; Fig. 7e). As with the other two dependent variables, reaction times in STEP were not associated with the maximum derivative PDR latency, $t(129.5) = 0.50$, $p = 0.6180$). Nor was there a significant association in the $REG_{10}$-$RAND_{20}$ condition ($t(115.6) = -0.20$, $p = 0.8415$)). However, RT in the $RAND_{20}$-$REG_{10}$ condition was associated with the maximum derivative PDR latency, $t(328.9) = 2.44$, $p = 0.0151$), and 1.8% of its variance.

In sum, the pattern of associations between participants' reaction times and various pupil measures suggests that the amplitude and timing of the pupillary response to both $RAND_{20}$-$REG_{10}$ and $REG_{10}$-$RAND_{20}$ are related to button press timing, but only to a modest degree, with RT accounting for between ~2 and 6% of estimated variance. While these analyses are limited by the relatively small amount of trials per condition/subject, this outcome suggests that the appearance of a PDR to the $RAND_{20}$-$REG_{10}$ transition in Exp3A is not primarily driven by the button press (or the decision to act, assuming the two are highly correlated[30]).

To confirm that the PDR to $RAND_{20}$-$REG_{10}$ observed in Exp3A is indeed not confounded by the motor response, we repeated the experiment using a delayed response paradigm (Exp3B; $n = 14$): Participants were instructed to monitor the tone sequences for pattern changes but indicate their response at the end of the trial. To control for vigilance and discourage participants from only attending to the beginning and end of a sequence, they were also instructed to monitor the stimuli for silent gaps, which could occur at any time (as in Exp1, 2, and 4; see Methods). The subset of sequences containing gaps or any button presses were excluded from the pupillometry analysis.

Behavioral results are summarized in Fig. 8a, b. For the change-detection task (Fig. 8a), hit and false alarm rate data demonstrated that performance on all transition conditions was at ceiling, with no difference between conditions ($F(1.99, 25.9) = 1.62$ $p = 0.216$). The false alarm effects observed in Exp3A were not seen here, likely because of the delayed response nature of the task. The gap detection data (Fig. 8b) also revealed no difference across conditions ($F(1.91, 24.8) = 2.86$ $p = 0.078$).

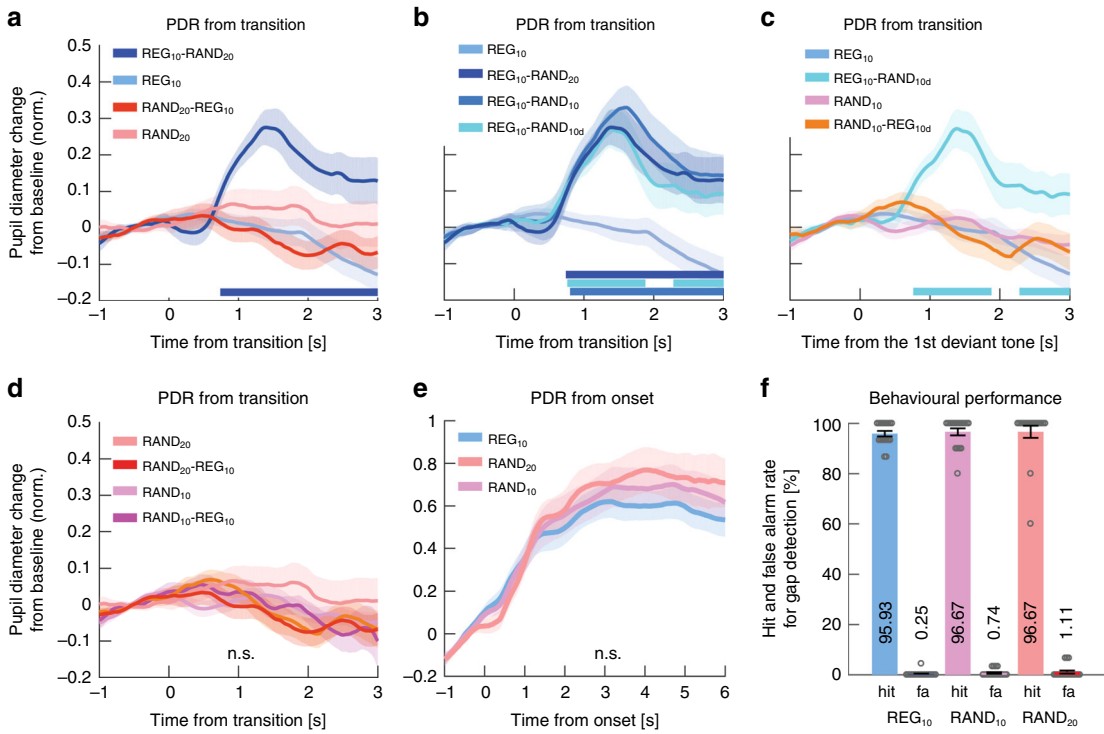

**Fig. 5** The PDR reflects sensitivity to 'pure' pattern violations. **a–d** Average pupil diameter over time relative to the transition in Experiment 2 ($n = 18$). Colored horizontal lines indicate time intervals where cluster-level statistics showed significant differences between each change condition and its no-change control. **a** The pupil responses to $REG_{10}$-$RAND_{20}$, $RAND_{20}$-$REG_{10}$ and their respective controls replicated the pattern observed in Experiment 1: $REG_{10}$-$RAND_{20}$ diverged from $REG_{10}$ at 740 ms post-transition; $RAND_{20}$-$REG_{10}$ did not differ statistically from $RAND_{20}$. **b** A significant PDR was observed for all conditions containing transitions from $REG_{10}$. The PDR in $REG_{10}$-$RAND_{10}$ increased from 420 ms, and statistically diverged from $REG_{10}$ 800 ms post-transition, peaking at 1600ms; the PDR to $REG_{10}$-$RAND_{10d}$ diverged from $REG_{10}$ at 760 ms up to 1880ms, peaking at 1380 ms. **c** A comparison between the two conditions which contained a change in alphabet at the transition ($REG_{10}$-$RAND_{10d}$, $RAND_{10}$-$REG_{10d}$). Only $REG_{10}$-$RAND_{10d}$ evoked a PDR. **d** None of the stimuli containing transitions from $RAND_{10}$ evoked a significant PDR. **e** Average pupil diameter over time, from stimulus onset. No differences were observed between any of the conditions. **f** Behavioral results for the gap detection task. Error bars are ± 1 SEM; gray circles represent individual participant data. Performance was at ceiling

The Pupillometry analysis results were likewise generally consistent with those observed in Exp3A: Fig. 8c plots the average pupil diameter relative to sequence onset. In line with Exp3A, and in contrast to Exp1 and 2, the sustained pupil diameter to $RAND_{20}$ rose above that to $REG_{10}$, likely reflecting the increased perceptual or computational demands associated with tracking random, relative to regular, patterns (see Discussion). However, these effects were somewhat more modest overall, including no difference between $REG_{10}$ and CONST, despite that seen in Exp3A. These moderate effects likely relate to the more relaxed tracking nature of the delayed response task, which may not have required the same level of vigilance and/or close tracking of sequence structure as that in Exp3A.

The PDRs, shown in Fig. 8d, were also reduced relative to those observed in Exp3A. Importantly, however, clear pupil dilations were observed in all three change conditions, including $RAND_{20}$-$REG_{10}$. Together with the results from Exp3A this confirms that the emergence of the PDR to $RAND_{20}$-$REG10$ is not a consequence of the motor act or decision to press the button. Rather, having listeners actively monitor and respond to the statistical transitions prompted a change in the underlying cognitive process, e.g. by delineating the category (or decision[30]) boundary between RAND and REG, and thereby rendering the transition as a model violation. We return to this point in the discussion.

**Exp4: pupil responses to transitions from randomness.** We have argued that to achieve effective model maintenance, the

brain must arbitrate between gradual and abrupt environmental changes. In other words, at each point in time, the brain must decide whether to continue updating its current representation of environmental contingencies or instead abandon the existing model and prioritize bottom-up evidence accumulation ("out with the old, in with the new"). Our results thus far suggest that what determines the difference between gradual and abrupt change can be gleaned through delineating the contingencies which evoke a PDR.

In REG-RAND, and trivially so in STEP, the statistical violation is immediately observable if listeners form a robust representation of the patterning of the REG sequences This could therefore be sufficient to trigger the abrupt-model violation signal.

By contrast, since absolutely any sequence of tones has the same probability under RAND, the detection of transitions out of this distribution is statistically more complicated. The lack of a PDR for $RAND_{20}$-$REG_{10}$ transitions, when not behaviorally relevant, is taken to indicate that that this transition is indeed not treated as an abrupt model violation. In Exp4 we explore whether the same is true for less (perceptually) complex regular patterns.

In Exp4A ($n = 12$) we asked whether the most basic regular pattern—a single repeating tone ($REG_1$)—evokes a PDR. Naive listeners (performing a gap detection task) were presented with $RAND_{20}$-$REG_1$ sequences (Fig. 9), in addition to STEP, $REG_{10}$-$RAND_{20}$ and $RAND_{20}$-$REG_{10}$. Replicating the result of Exp1, we observed a PDR to $REG_{10}$-$RAND_{20}$ but not to $RAND_{20}$-$REG_{10}$. By contrast, the $RAND_{20}$-$REG_1$ transition evoked a fast-onset and

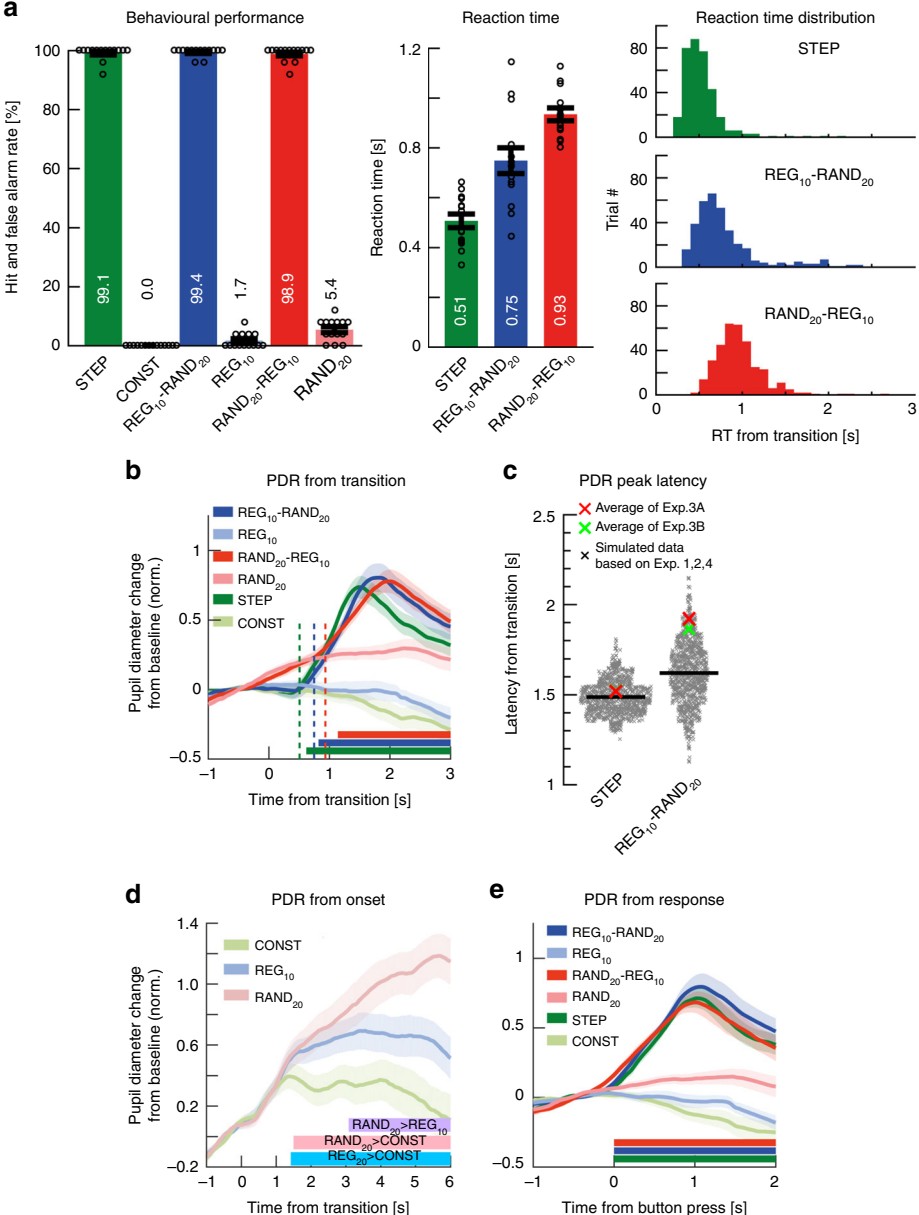

**Fig. 6** Experiment 3A ($n = 14$): Active transition detection. **a** Behavioral performance (transition detection). [Left] Hit rates and false positive rates. Circles indicate individual subject data; error bars are ±1 SEM. [Middle] Reaction times (RT). [Right] The distribution of RTs (across trials and participants) for each transition condition. The variance of the RT distribution of $RAND_{20}$-$REG_{10}$ was smaller than that of $REG_{10}$-$RAND_{20}$ (Levene's test, $F(1,690) = 14.426$, $p = 0.0002$). **b** Average pupil diameter relative to the transition. Solid lines represent the average normalized pupil diameter, relative to the transition. Shading shows ±1 SEM. Colored horizontal lines indicate time intervals where cluster-level statistics showed significant differences between each change condition and its control. Dashed vertical lines mark the average RT for each condition. Clear PDRs were observed for all three transitions. The PDR to STEP increased from ~490 ms post-transition, peaking at 1500 ms; it statistically diverged from CONST from 620 ms through to sequence offset. For $REG_{10}$-$RAND_{20}$, the responses commenced ~750 ms post-transition, peaking at 1800 ms, and statistically diverged from REG at 820 ms post-transition through to sequence offset. For $RAND_{20}$-$REG_{10}$, the response started at 930 ms, peaked at 1980 ms, and statistically diverged from its control $RAND_{20}$ from 1140 ms onwards. **c** A comparison of PDR peak latency to STEP and $REG_{10}$-$RAND_{20}$ in Experiment 3 relative to Experiments 1, 2, and 4 (see Methods). The scatterplots show the distribution of the simulated peak latency of STEP (left) and $REG_{10}$-$RAND_{20}$ (right) in the gap detection experiments. The red and green crosses indicate the mean peak latency in Experiments 3A and 3B (see below), respectively. The results showed no difference for STEP (Exp 3A: mean = 1520 ms, $p = 0.338$, Expt3B: mean = 1521 ms, $p = 0.341$), but a greater latency for $REG_{10}$-$RAND_{20}$ in Exp 3A (mean = 1921 ms, $p = 0.048$, Exp 3B: mean = 1865 ms, $p = 0.090$). **d** Average pupil diameter over time relative to the sequence onset. Colored lines indicate time intervals where cluster-level statistics showed significant differences between conditions. $RAND_{20}$ and $REG_{10}$ statistically diverged from CONST at 1500 and 1420 ms post-onset, respectively. Interestingly, $RAND_{20}$ evoked a larger tonic PDR than $REG_{10}$ from 3080 ms post-onset. **e** Average pupil diameter over time, relative to button press

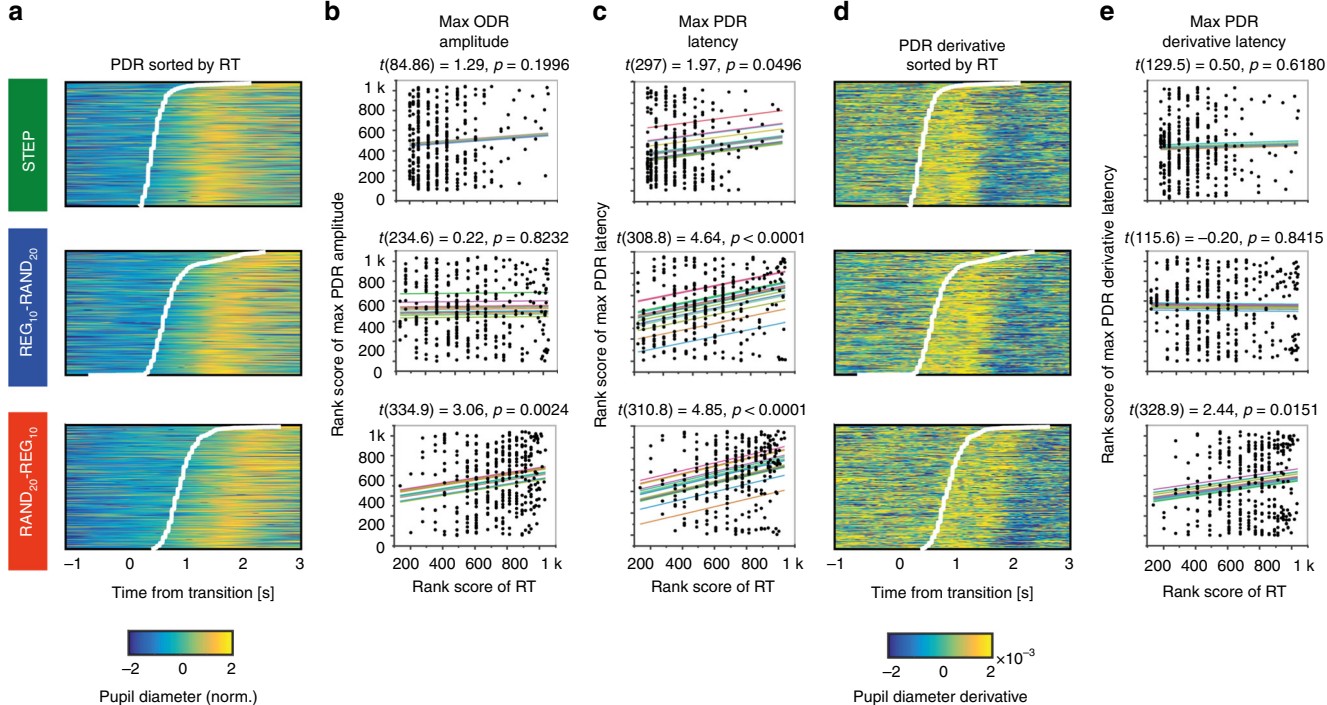

**Fig. 7** Relationship between RT and the pupil dilation response. **a** Single trials sorted by RT (*y*-axis, RT indicated by white lines) shown against the time relative to the transition (*x*-axis) with the colors showing pupil diameter (the warmer the color, the larger the pupil). **b** Scatter plots show the maximum PDR amplitude for each trial (ranked low to high on the *y*-axis) versus ranked RT (smaller values faster) for the same trial (*x*-axis), separated by condition as in **a**. Each line of fit shows the modeled random effect of subject (offset), with slope the fixed effect of RT. Fixed-effects *t*-values and associated *p*-values appear above each fitted scatterplot. **c** Rank maximum PDR latency vs. RT. Scatterplot and fitting as in **b**. **d** Single trials sorted by RT (*y*-axis, RT indicated by white lines) shown against the time relative to the transition (*x*-axis) with the colors showing the rate of change of pupil diameter (the warmer the color, the larger the rate of change in pupil size). **e** Rank maximum derivative latency vs rank RT (scatterplot fits as in **c**)

robust PDR of similar amplitude to that evoked by transitions from regularity to randomness. This result is consistent with previous demonstrations that the violation of randomness by repetition evokes an MMN-like response[32,33] —a finding which was taken to suggest that the auditory system represents stochastic frequency variation as a regularity per se[34].

As with the previous experiments, there was no significant difference between REG$_{10}$ and RAND$_{20}$ pre-transition (Fig. 10b), and no behavioral difference in the gap detection task across the three conditions (Fig. 10c).

How does the PDR evolve as the regularity becomes more complex, e.g., as we add more elements to the regular pattern? In Exp4B, in addition to RAND$_{20}$-REG$_1$, we also included transitions from RAND$_{20}$ to repeating 2-, 5-, and 10-tone patterns (REG$_2$, REG$_5$, REG$_{10}$, Fig. 9; see also Supplementary Fig. 2 for information theoretic modeling of the sequences).

Due to the increased number of conditions and therefore longer experiment time, data were noisier than in the previous experiments. Thus, to replicate the effects from the first group (group A; *n* = 15), the experiment was repeated in another group of listeners (group B; *n* = 15). We also collapsed the data across both groups to maximize statistical power.

For both groups, the behavioral performance in the gap detection task was at ceiling (Fig. 10e). Figure 10d plots the results of Exp4B for each group separately, and when pooled together. In each group, and as in Exp4A, RAND$_{20}$-REG$_1$ evoked a significant PDR; The average pupil diameter in RAND$_{20}$-REG$_2$ and RAND$_{20}$-REG$_5$ appeared to suggest a small gradual increase between 640 ms and 800 ms. This effect reached significance, for RAND$_{20}$-REG$_2$ only, in group B. Collapsing the data across groups confirmed a significant PDR for RAND$_{20}$-REG$_2$ between

840 and 1560 ms post-transition. The PDR for RAND$_{20}$-REG$_5$ remained non-significant. These effects are also mirrored in the PD rate analysis (see Supplementary Fig. 3). As in all previous experiments, no PDR was observed for RAND$_{20}$-REG$_{10}$.

Overall the results of Exp4B demonstrate a sharply reduced PDR for regularities more complex than REG$_1$: while REG$_1$ consistently evoked a robust response in both groups A and B, and with an amplitude identical to that observed for STEP and REG$_{10}$-RAND$_{20}$ (Exp4A, Fig. 10a), the PDR to RAND$_{20}$-REG$_2$ was substantially reduced. The small effect of REG$_2$ may indicate that for some subjects, or in a subset of trials, a PDR was present. The lack of a PDR for more complex regularities suggests that the associated transitions are treated as a gradual rather than abrupt transition with respect to the internal model maintained for RAND$_{20}$.

## Discussion

Research in animal models has established a robust link between phasic pupil responses and spiking activity within the LC, providing compelling evidence for pupil dynamics as an indirect measure of NE release[25]. Our observations are therefore interpreted in the context of understanding the role of the pupil-linked LC-NE system in reporting on aspects of the statistics of rapid sensory signals.

A large body of work has suggested a gating role for NE in balancing bottom-up-driven sensory processing vs. top-down priors[4,12–15,35–43]. Indirect measures of NE release, based on pupillometry, have also revealed an association between NE signaling and increased learning rates – a proxy for model resetting[2,11]. However, the existing literature is limited by the

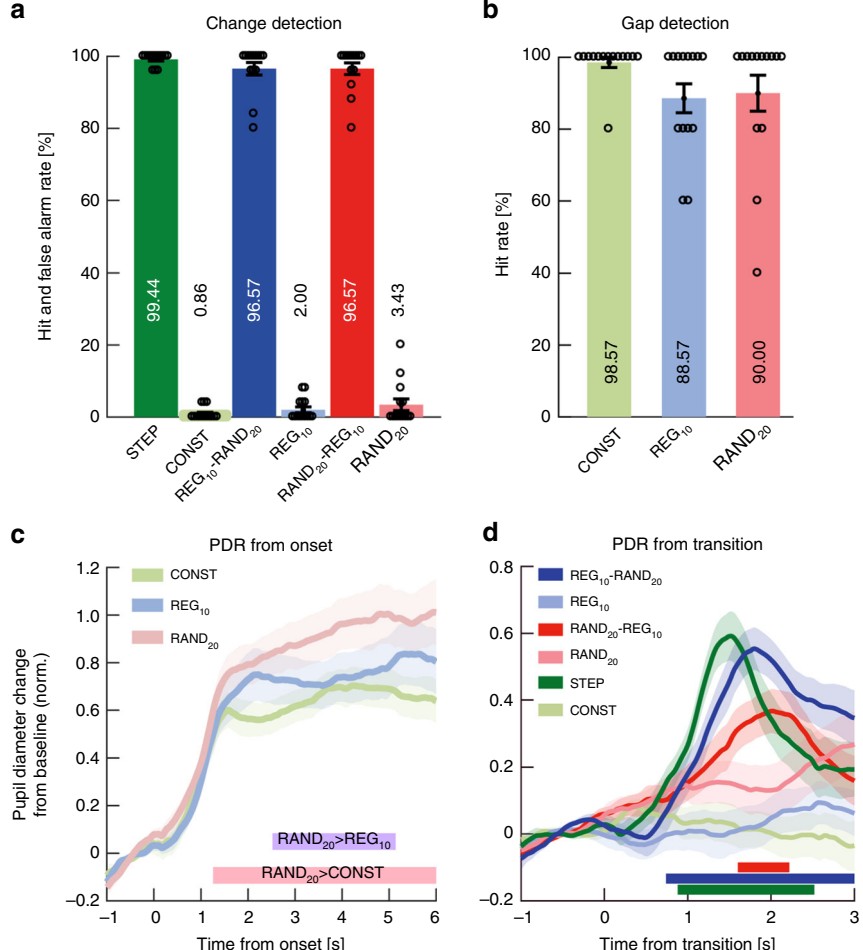

**Fig. 8** Experiment 3B ($n = 14$): Active transition detection (delayed response). **a** Transition detection task: Hit rates and false positive rates. Circles indicate individual subject data; error bars are ±1 SEM. **b** Gap detection task. **c** Average pupil diameter over time relative to sequence onset. Colored lines indicate time intervals where cluster-level statistics showed significant differences between conditions. $RAND_{20}$ statistically diverged from CONST at 1260 ms through to sequence offset and from $REG_{10}$ between 2520 ms and 5120 ms post-onset. **d** Average pupil diameter relative to the transition. Shading shows ±1 SEM. Colored horizontal lines indicate time intervals where cluster-level statistics showed significant differences between each change condition and its control. The PDR to STEP increased from ~200 ms post-transition, peaking at 1520 ms; it statistically diverged from CONST between 880 and 2520 ms. For $REG_{10}-RAND_{20}$, the responses commenced ~500 ms post-transition, peaking at 1800ms, and statistically diverged from REG at 740 ms post-transition through to sequence offset. For $RAND_{20}-REG_{10}$, the response rose at 1000 ms, peaked at 2020 ms, and statistically diverged from its control $RAND_{20}$ between 1600 and 2220 ms

fact that most of the experimental results taken to support the 'NE as an interrupt signal' hypothesis have involved tasks in which inputs evolve slowly over time and participants either make active decisions about stimulus predictability or are required to form stimulus–response associations[2,4,10,11,20,21,29]. In contrast, here we used rapid, not explicitly trackable[44], auditory patterns, specifically structured to evoke model resetting vs. model updating. We demonstrate that when pattern changes were behaviorally irrelevant, pupil dilation responses (PDR) were evoked exclusively by changes associated with violations of regularity. Thus, we extend to the pre-attentive case findings about NE and interrupts originally derived from decision-making tasks and demonstrate that the pupil-linked LC-NE system plays an obligatory role in tracking the statistics of unfolding sensory input.

We further demonstrate that behavioral relevance exerted a major effect on pupil dynamics, changing the responses both during the establishment of patterns, and at transition periods. Overall, the results are consistent with a hypothesized role of NE as a model interrupt signal, and provide a rich view of the

contingencies that have automatic and/or controlled access to this interrupt.

Deviants or salient changes in sound sequences are well known to evoke PDRs, even under passive listening conditions[45,46]. These observations have prompted a suggestion that, as part of a broader fight-or-flight response, pupil activation reflects the operation of an interrupt signal that halts current ongoing activities to allow an attentional shift towards the new event, thus facilitating adaptive behavior[46,47]. Phasic LC-NE activation has duly been hypothesized to serve as a neural interrupt signal for unexpected events[10,12,14], prompting the resetting of top-down connectivity when sensory information indicates that the currently instantiated model of the environment is no longer valid. These observations raise the obvious – and behaviorally important – question of what exact changes are able to drive the interrupt signal.

A common idea is that sensory processing involves rich statistical modeling[48–50], which suggests that many changes might drive the interrupt signal. However, not only are there obvious dangers to making interruption too promiscuous when neural

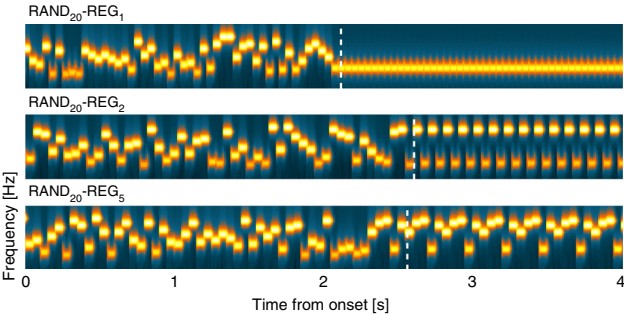

**Fig. 9** Example spectrograms for the additional stimuli used in Exp 4A, B. $RAND_{20}$-$REG_1$ consisted of a transition from a random sequence (generated by sampling frequencies from the full pool with replacement; $RAND_{20}$) to a single repeating tone. $RAND_{20}$-$REG_2$ consisted of a transition from a $RAND_{20}$ sequence to a regular pattern consisting of two randomly selected tones. $RAND_{20}$-$REG_5$, consisted of a transition from a $RAND_{20}$ sequence to a regular repeating pattern consisting of 5 tones. Dashed vertical white lines indicate the transition time, defined as occurring after the first full regularity cycle. See also Supplementary Fig. 2 for an information theoretic[28] characterization of the stimuli: All transition conditions show a gradual decline in information content (IC) over several tones as the model discovers the predictable structure within the sequence. For presentation purposes the plotted sequence lengths are equal. Actual durations varied randomly between 6 and 7.5 s. In Experiment 4A the stimulus set also contained $RAND_{20}$-$REG_{10}$, $REG_{10}$-$RAND_{20}$, STEP, $REG_{10}$, $RAND_{20}$ and CONST sequences. In Experiment 4B the stimulus set additionally contained $RAND_{20}$-$REG_{10}$ and $RAND_{20}$ sequences

processing is focusing on a task for which the statistics are irrelevant, but it is also computationally costly to build detailed models of complex signals when these do not matter. This consideration motivated a heuristic separation between expected and unexpected uncertainty[12,15] with events falling in the latter category triggering model interruption, allowing a more sophisticated model-building process to occur, if appropriate.

Here, we exploited the statistical asymmetry between REG-RAND and RAND-REG to gain further insight into these limits. Both transitions are equally behaviorally detectable, but differ in terms of their underlying statistical structure: The transition in REG-RAND is computationally simple to detect, given knowledge of REG. However, for the RAND-REG transition, the RAND model is not directly falsified because all tones within the REG pattern are strictly consistent with the RAND model. The hypothesis is therefore that they are treated in terms of expected – rather than unexpected – uncertainty, leading to gradual rather than abrupt model change (Fig. 1, Supplementary Fig. 1). In line with the formulation proposed by Dayan & Yu[12], we found that, when the transition was irrelevant to the behavioral goal, the LC-NE system appeared to ignore RAND-REG transitions. That was observed for even moderately complex REG patterns, not only refutes the suggestion that any perceptually salient and contextually novel set of observations can drive NE, but also hints at limits to the statistical model-building process.

To tap the statistical model building process for $RAND_{20}$, we systematically investigated $RAND_{20}$-REG transitions by modulating REG cycle length (Exp4). A default hypothesis was that since all have similar statistical structure (a smooth change in information content; Supplementary Fig. 2), they should all result in effects identical to that observed for $RAND_{20}$-$REG_{10}$ - namely not evoke a PDR. We indeed observed a reduced PDR for $RAND_{20}$-$REG_2$ onwards which we interpret as further evidence for the specificity of the pupillary response.

Interestingly, the $RAND_{20}$-$REG_1$ transition did evoke a PDR. This may be taken to indicate that, unlike for REG with longer

cycles, a transition to $REG_1$ is treated as an abrupt model violation with respect to the internal model maintained for $RAND_{20}$. One suggestion is that the brain may engage in a form of automatic latent model building using just the last few tones. If the latent model based on those few tones fits them much better than the prevailing model, then an abrupt change is reported. Under this hypothesis, the fact that even as simple a sequence as two alternating tones does not generally lead to model change-detection suggests stringent constraints on the automatic model construction – perhaps that it encompasses no more than two successive tones. It is tempting to speculate that such model construction could be implemented by low-level coding mechanisms e.g. adaptation or repetition suppression, both of which would lead to detectably unusual patterns of activity in tonotopically organized neural populations.

Overall, the results indicate that the presence of a PDR as a marker for unexpected uncertainty can be used to probe the observers automatically construct of their surroundings. The demonstration of a distinct boundary between expected and unexpected uncertainty under behaviorally irrelevant listening conditions calls for future modeling and experimental work to outline the properties of this distinction and its implications for perception.

The PDR specificity is consistent with patterns of brain responses measured with Electro- and Magnetoencephalography in naïve, distracted listeners (Fig. 1b). Robust brain responses are observed to both RAND-REG and REG-RAND transitions, but importantly, the response dynamics are distinct, revealing the differing computational demands of each transition type: RAND-REG transitions evoke a progressive increase in sustained brain responses, hypothesized to reflect the gradual increase in model precision associated with the increased predictability of the REG patterns[22]. This is underpinned by a distributed brain network of auditory cortical, frontal and hippocampal sources[22,24]. Together, these sources are hypothesized to support the instantiation of a top-down model, producing increasingly reliable priors for upcoming sounds[51]. In contrast, REG-RAND transitions evoke a mismatch response (similar in its dynamics to the MMN[52]), followed by an abrupt drop in sustained activity, in line with immediate suppression of top-down prior expectations[22]. The activity then settles at a low sustained level, consistent with the weaker statistical constraints in the RAND pattern.

The PDR results point to a potential role for NE signaling in supporting the Electro/Magnetoencephalography indexed resetting response observed during REG-RAND transitions. The relevant neural circuit may involve signaling from MMN-related brain systems (Auditory Cortex and right IFG[53]) to the LC-NE system, possibly via the ACC[10,40,54–56] or orbitofrontal cortex[6,57]. LC activation would then trigger NE-mediated rapid interruption of the temporo-frontal network associated with generating top-down prior expectations. Further investigation combining pupillometry and sensitive source imaging are necessary to identify these circuits, and to test the proposed linked between the MMN response and NE release.

Importantly, the presence of extensive brain activation to behaviorally irrelevant RAND-REG transitions, suggests that the lack of a PDR in that condition is not due to those transitions not being detected in the passively listening brain. Rather, it appears that this information does not modulate activity in the LC-NE system.

We demonstrated that the PDR specificity which is observed under passive listening conditions can be reversed when transitions are behaviorally relevant. Rendering the pattern changes behaviorally relevant resulted in marked differences in pupil response dynamics. Most notably, active monitoring gave rise to a PDR to $RAND_{20}$-$REG_{10}$ transitions. These effects were not

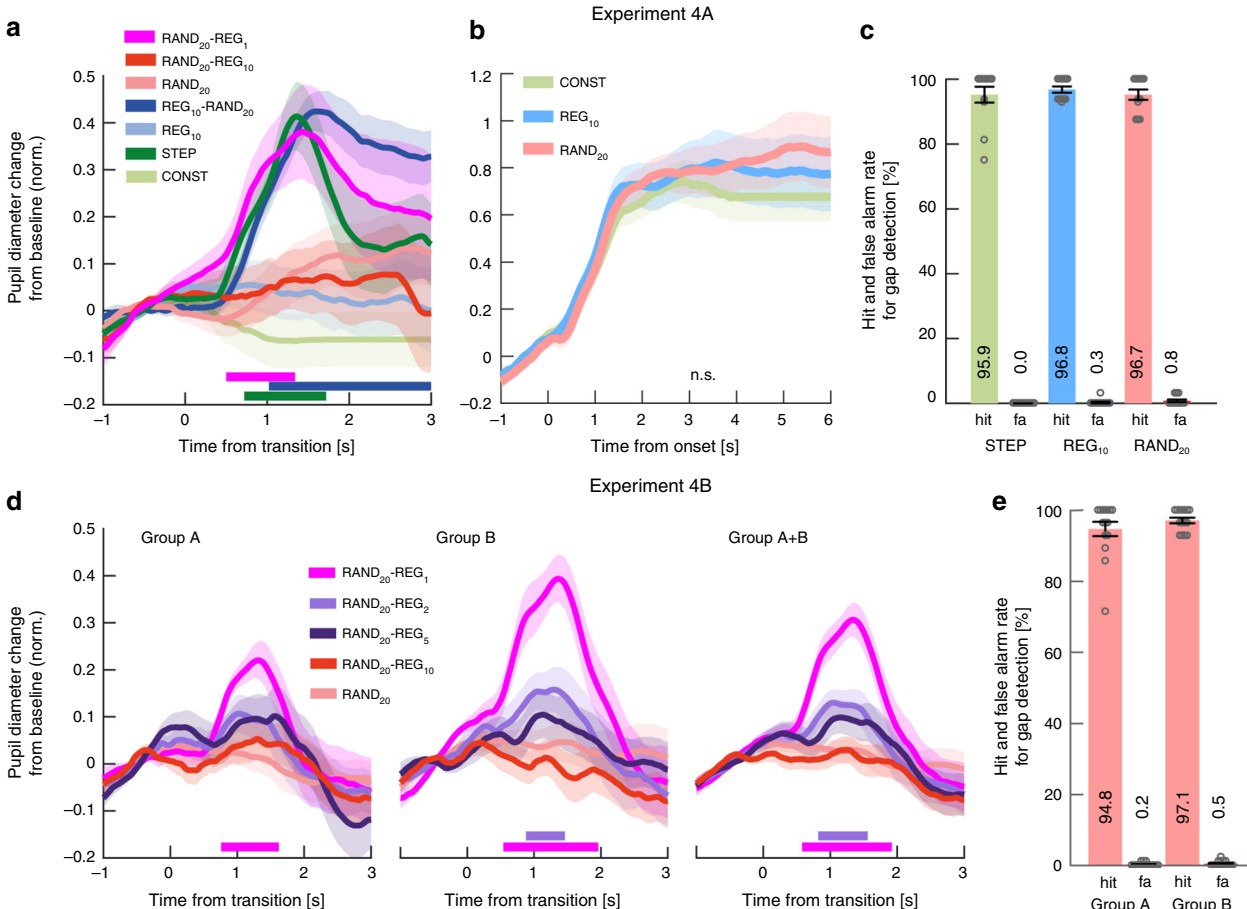

**Fig. 10** Abrupt reduction in the PDR for regularities more complex than REG$_1$. **a** Average pupil diameter relative to the transition in Experiment 4A ($n = 12$). Solid lines represent the average normalized pupil diameter, relative to the transition. The shaded area shows ±1 SEM. Colored horizontal lines indicate time intervals where cluster-level statistics showed significant differences between each change condition and its control. A robust PDR was evoked by the transition in RAND$_{20}$-REG$_1$, becoming significant between 500 and 1340 ms post-transition The data also replicate the general pattern in Experiments 1 and 2: Both STEP and REG$_{10}$-RAND$_{20}$ evoked a PDR; the former started from 720 ms lasting through to 1720ms, and the latter from 1020 ms onwards. No significant difference between RAND$_{20}$-REG$_{10}$ and RAND$_{20}$ was observed. **b** Average pupil diameter over time from stimulus onset. No differences were observed between any of the conditions. **c** Behavioral results for the gap detection task in Experiment 4A with ±1 SEM error bars, and gray circles representing individual participant data. There was no statistical difference between conditions. **d** Average pupil diameter over time relative to the transition in Experiment 4B: [Left] Group A ($n = 15$). A clear PDR is observed for RAND$_{20}$-REG$_1$ which diverged from its control, RAND$_{20}$, from 746 to 1620ms no significant differences were observed in the other conditions. [Middle] Group B ($n = 15$). RAND$_{20}$-REG$_1$ diverged from RAND$_{20}$ between 540 ms and1960 ms. RAND$_{20}$-REG$_2$ also showed a significant PDR between 880ms-1460ms, peaking at 1300 ms. RAND$_{20}$-REG$_5$ and RAND$_{20}$-REG$_{10}$ were not significantly different from RAND$_{20}$. Comparing between transition conditions, RAND20-REG1 was significantly greater than the other transition conditions from ~600 ms to ~1800ms post-transition. [Right] Both groups combined ($n = 30$). Significant PDRs were observed for RAND$_{20}$-REG$_1$ (from 580 to 1920ms) and RAND$_{20}$-REG$_2$ (from 820 to 1560 ms). Comparing between transition conditions, RAND20-REG1 was significantly greater than the other transition conditions from ~600 ms to ~1800ms post-transition. **e** The behavioral performance for both groups was at ceiling

strongly linked to the execution of a motor command or the decision to respond, as evidenced by the fact that RT accounted for relatively little variance in various PDR metrics and that the effects were largely preserved in a delayed response version of the task. Therefore, these behavior-related changes in the PDR likely reflect a change in the functional state of the LC-NE system, or inputs to it. For example, it is possible that task relevance, or heightened arousal under behaviorally relevant conditions, leads to a richer representation of the statistics of the RAND patterns or contributes to the emergence of a category boundary between REG and RAND patterns thereby rendering the transitions, in both directions, as model violations. Alternatively, behavioral relevance may alter the boundary between 'expected' and 'unexpected' uncertainty, resulting in a threshold change for model reset.

We also observed a change in the dynamics of tonic pupil activity, i.e. in response to the ongoing sequence before the transition. When RAND and REG states were behaviorally irrelevant (in all but the third experiment), we observed no difference between the ongoing response to REG and RAND throughout the entire epoch (Figs. 2c, d, 5e, and 10b). However, making transitions between these states behaviorally relevant resulted in diverging PDRs to the different conditions themselves, even in the absence of a transition (Fig. 6b, d). Notably, these differences were observed even though the statistical structure *per se* is not explicitly trackable by listeners, due to the rapid rate at which successive tones are presented. Previous work has linked tonic pupil diameter differences to representation of expected uncertainty[2,11] possibly driven by cholinergic signaling[15,26]. The present effects may be consistent with this interpretation; as

indeed $RAND_{20}$ is associated with less reliable priors than $REG_{10}$. However, the fact that these differences in pupil diameter were observed exclusively during the active change-detection task must therefore suggest that cholinergic activation is dependent on behavioral relevance and is not involved in automatic tracking of sequence predictability. An alternative, but not mutually exclusive, possibility is that this effect may reflect heightened vigilance or listening effort[58] arising through active sequence structure scanning, which is more demanding for $RAND_{20}$[24].

In conclusion, the data reported here demonstrate that the pupil-linked LC-NE system plays an obligatory role in tracking and evaluating the statistics of unfolding sensory input, thereby supporting brain networks involved in maintaining flexible perceptual representations in changing environments. However, this system is confined in the circumstances under which it signals an interrupt, particularly in the absence of a remit from a task. Together with previous work in the decision-making and learning fields, the present results establish a unified view of NE as a model interrupt signal operating on multiple time scales, from those relevant to tracking reward environments in the context of decision making to tracking rapidly unfolding sensory environments during perception.

## Methods

The experimental procedures were approved by the Research Ethics Committee of University College London. Participants were provided written informed consent and were paid for their participation.

**Participant details**. All participants reported normal hearing, normal or corrected-to-normal vision, and no history of neurological disorders. The following exclusionary criteria were consistently applied across all experiments: To ensure that observed changes in pupil diameter were not blink-related artifacts, participants were excluded if they blinked in more than 50% of trials. Additionally, participants were excluded if their mean gaze location exceeded three standard deviations from the group mean.

Experiment 1A: data from 18 participants (11 females; aged 20–29, average 23.41) are presented. Data from one additional participant were excluded due to failure to complete the experiment. Two further participants were excluded due to high blink rates in the STEP condition.

Experiment 1B: data from 14 new participants (13 females; aged 22–26, average 23.1) were used in the analysis. Five additional participants were excluded due to high blinks rates. One further participant was excluded due to poor behavioral performance (0% gap detection hit rate in REG sequences).

Experiment 2: data from 18 new participants (15 females; aged 20–35, average 25.1) are reported. Two additional participants were excluded: one due to high blink rates, and one due to wandering gaze.

Experiment 3A: data from 14 participants (10 females; aged 22–30, average 24.3) are presented. None were excluded.

Experiment 3B: data from 14 participants (12 females; aged 20–31, average 23.9) are presented. None were excluded.

Experiment 4A: data from 12 new participants (9 females; aged 21–26, average 23.6) are presented. None were excluded.

Experiment 4B: this experiment was performed twice; a total of 30 new participants took part, with 15 participants initially (11 females; aged 20–29, average 23.5) and a new group of 15 participants subsequently (14 females; aged 20–25, average 22.5) to replicate the results of the first cohort. None were excluded.

As a confirmation of the adequacy of sample-sizes, we quantified effect sizes in Expt1A and Exp1B, which were chronologically the first and second experiments in this project, and used those values for a power analysis to confirm that N in the subsequent experiments (2–4) was adequate. The power analysis was conducted in the G*Power software package[59], with the following settings: $1 - \beta > 0.8$, and $p = 0.05$.

To produce a measure of the magnitude of the PDR effect in Exp1A, B, we first found the latency of the PDR peak in the grand average for STEP and REG-RAND respectively (Fig. 2a, b), and used this value to obtain, for each subject, the amplitude in the transition conditions and their respective controls. A net effect measure (quantifying the size of the PDR) was then computed for each subject by taking the difference between each transition and its control (no-change) condition. Note that this manner of reducing the PDR effect to a single number per subject is necessarily more conservative than the time-sensitive analysis employed to quantify the PDR in the main analysis. Both Exp1A and Exp1B enjoyed large effects sizes: Cohen's $d\sim1.2$–1.7 for STEP and $d = 0.7$–0.8 for RAND-REG. An overall estimated effect size of 0.8 was therefore fed into the power analysis and yielded an $n = 12$, confirming that all further experiments (Experiments 2–4; $N > = 12$ for all) were adequately powered.

**Pupil size measurement and analysis**. Participants sat in front of a monitor at a viewing distance of 60 cm in a dimly lit, acoustically shielded room (IAC triple walled sound-attenuating booth), with their head supported on a chinrest. They were instructed to continuously fixate at a white cross presented at the center of the screen (BENQ XL2420T with resolution of 1920x1080; refresh rate of 60 Hz) against a black background. The visual display remained constant throughout the session. An infrared eye-tracking camera (Eyelink 1000 Desktop Mount, SR Research Ltd.), positioned just below the monitor, continuously tracked gaze position and recorded pupil diameters, focusing binocularly at a sampling rate of 1000 Hz. The standard five-point calibration procedure for the Eyelink system was conducted prior to each experimental block. Participants were instructed to blink naturally.

Only the left eye was analyzed. To avoid contamination by blinks, which tended to increase towards the end of the stimulus, the final 0.5 s of each trial were cut from the analysis. The epochs therefore spanned one second before to two seconds post transition (Experiment 1) or three seconds post transition (all other experiments). This cut-off was comfortably beyond the time needed to detect the transitions, as corroborated by behavioral and MEG results (Experiment 3 and the previous MEG work[22]). Intervals where the eye tracker detected full or partial eye closure were automatically treated as missing data and recovered with shape-preserving piecewise cubic interpolation. Data were then smoothed with a 150-ms Hanning window. Epochs with more than 50% missing data were excluded from the analysis (<2 trials per subject). To allow for comparison across trials, subjects and experiments, data for each subject in each block were z-normalized based on the mean and standard deviation computed across all the data (all epochs, all conditions) within the block. To remove variance related to the pre-transition baseline, baseline pupil size (mean value over the 1-second pre-transition interval) was regressed out in a point-by-point manner from the data for each stimulus condition in each subject (regression coefficients were computed independently for each sample point in each condition). Thereafter, the obtained time series (residuals after accounting for linear baseline dependences) were time-domain-averaged across all epochs of each condition type to produce a single time series for each condition. Matched no-transition conditions were processed in a similar manner around dummy transition times set to match those in the transition conditions.

To compare pupil dynamics from sequence onset, the data in the no-change conditions (REG, RAND, CONST) were epoched from 1 s before sequence onset to 6 s post-onset and baseline-corrected by subtracting the 1 s pre-onset interval.

**Pupil event rate analysis**. Pupil event rate analysis compared the incidence of pupil dilation or constriction events. Following Joshi et al.[25], events were defined as local minima (dilations; PD) or local maxima (constrictions; PC) with the constraint that continuous dilation or constriction is maintained for at least 75 ms (yellow dots in Fig. 3) or 300 ms (black dots in Fig. 3). Both thresholds provided consistent data (see results), as did intermediate thresholds. The rate was estimated for each subject separately by using a sliding 500 ms window over all trials in each condition and comparing rate changes across time and condition (see Statistical Analysis below). This relatively long window enabled us to capture possible subtle changes in the rate of occurrence of PD events. The analysis interval was between 2 s before to 2 s after the transition. Previous work[22] demonstrated that brain responses to the transitions occur within <300 ms and behavioral responses (button press) are completed by 1000 ms (see also Exp3 below), thus suggesting that the analysis interval is appropriate for revealing any effects. We also estimated rate by tallying PD or PC events with non-overlapping 500 ms windows, and by convolving with an impulse function[25,60]. For each condition, in each participant and trial, the event time series were summed and normalized by the number of trials and the sampling rate. Then, a causal smoothing kernel $\omega(\tau) = \alpha^2 \times \tau \times e^{-\alpha\tau}$ was applied with a decay parameter of $\alpha = \frac{1}{50}$ ms[60–62]. All analyses yielded identical results, therefore only the former is reported.

**Experiment 1A Stimuli and procedure**. The stimuli were sequences of concatenated tone-pips (50 ms) with frequencies drawn from a pool of 20 fixed values (log-spaced) between 200 and 2000 Hz. The tone-pips were arranged according to six frequency patterns, generated anew for each participant (Fig. 1): CONST sequences consisted of a single repeating tone, chosen by randomly selecting a frequency from the pool on each trial; STEP sequences consisted of a step change from one repeating tone to another repeating tone of a different frequency (both frequencies randomly drawn on each trial); $REG_{10}$ sequences were generated by randomly selecting (with replacement) 10 frequencies from the pool and then iterating that sequence to create a regularly repeating pattern (with new patterns generated on each trial); $RAND_{20}$ sequences were generated by randomly sampling frequencies from the pool with replacement; $REG_{10}$-$RAND_{20}$ and $RAND_{20}$-$REG_{10}$ sequences contained a transition between a regular pattern and a random pattern. The stimulus length varied randomly between 5 and 7 s, with a jittered transition time at around 2.5- and 3.5-seconds post-onset.

Sounds were presented diotically through headphones (Sennheiser HD558) via a Creative Sound Blaster X-Fi sound card (Creative Technology, Ltd.) at a comfortable listening level, self-adjusted by each participant. Stimulus presentation and response recording were controlled with the Psychtoolbox package (Psychophysics Toolbox Version 3[63]) in MATLAB (The MathWorks, Inc.).

146 stimuli – 24 for each condition - were randomly presented in four consecutive blocks (separated by 3 min breaks) with an inter-trial interval of six seconds. In all, 25% of the signals contained a silent gap, occurring at any time from 250 ms post onset to 750 ms pre-offset. Participants were instructed to monitor the sequences for these events and to respond by pressing a button as quickly as possible. To equate for task difficulty, the gap consisted of one missing tone (50 ms) in the CONST and STEP sequences, and two missing tones (100 ms) in REG and RAND sequences. Visual feedback, lasting 400 ms, was provided immediately at the end of each sequence. Trials containing a gap and trials in which participants made a false positive were excluded from further analysis.

**Experiment 1B stimuli and procedure**. The stimuli and procedure were identical to Experiment 1A, except that only three blocks were run, for a total of 108 stimuli. The data from Experiment 1A indicated that this was sufficient to measure the relevant effects. To address behavioral differences between conditions observed in Experiment 1A, the gap in RAND sequences was lengthened to 3 tones (150 ms).

**Experiment 2 stimuli and procedure**. The stimulus set is described in Fig. 4. Stimulus length was randomly varied between 6 and 8 s, with the transition jittered between 3 and 4 s after sequence onset. A total of 240 stimuli were presented in random order over 5 consecutive blocks: 60 $REG_{10}$, 20 $REG_{10}$-$RAND_{10}$, 20 $REG_{10}$-$RAND_{10d}$, 20 $REG_{10}$-$RAND_{20}$, 40 $RAND_{10}$, 20 $RAND_{10}$-$REG_{10}$, 20 $RAND_{10}$-$REG_{10d}$, 20 $RAND_{20}$, and 20 $RAND_{20}$-$REG_{10}$. In all, 20% of the sequences contained a gap, with equal probability spread across conditions. Gap lengths were as in Experiment 1B.

**Experiment 3A stimuli and procedure**. The stimulus set was identical to that in Experiment 1 but stimuli contained no gaps. Participants were instructed to press space bar as quickly as possible after detecting a pattern change in the sound sequence. Key presses were checked after each tone, so the resolution of the reaction time measurement was 50 ms. In total, 150 stimuli were presented in random order over 5 consecutive blocks – 25 of each condition – with an inter-trial interval of five to seven seconds.

**Experiment 3B stimuli and procedure**. The stimulus set was identical to that in Experiment 1B. In total, 180 stimuli (30 of each condition) were presented in random order over 5 consecutive blocks. 16.7% of the signals contained a silent gap, the length of which was as in Experiment 1B.

Participants were instructed to monitor the sequences for pattern changes (as in Experiment 3A) and were queried about the presence of a change at the end of the trial (delayed response paradigm). Simultaneously, they also monitored the sequence for silent gaps (as in Experiments 1, 2, and 4) and were instructed to respond as quickly as possible, by pressing the space bar. After the tone-pip sequence has ended, the white fixation cross changed to a white question mark to indicate the response interval. Participants specified whether the sequence contained a change ('Yes' or 'No') by pressing one of two buttons (counterbalanced across participants). Following the change-detection response, visual feedback for both gap detection and change detection was given at the center of the screen (1 s). The next trial began following a 5-s inter-trial interval. Participants provided the change-detection responses with their dominant hand, and the gap detection with the other hand. Each block was ~8–9 min long, followed by an 8 min break. Trials containing a gap, trials during which participants pressed any key during the sound presentation, or trials with an incorrect change-detection response were excluded from the analysis of the pupillometry data.

**Experiment 4A stimuli and procedure**. The stimulus set consisted of the conditions used in Experiment 1 (Fig. 1) and additionally included a new condition: $RAND_{20}$-$REG_1$ – which consisted of a transition from a random sequence to a sequence of fixed frequency tones ($REG_1$) (see Fig. 9). In total, 288 trials were presented in random order over 8 consecutive blocks: 24 $RAND_{20}$-$REG_1$, 24 $RAND_{20}$-$REG_{10}$, 48 $RAND_{20}$, 48 $REG_{10}$, 48 $REG_{10}$-$RAND_{20}$, 48 CONT and 48 STEP, with one-third of the sequences containing a gap (lengths as in Experiment 1B).

**Experiment 4B stimuli and procedure**. The stimulus set was expanded to include two additional conditions: $RAND_{20}$-$REG_2$ consisted of a transition from a random sequence to a sequence to a regular pattern of two alternating tones; $RAND_{20}$-$REG_5$ consisted of a transition from a random sequence to a sequence to a regular pattern of five alternating tones (Fig. 9). Overall, 168 stimuli were presented in random order over 7 consecutive blocks including 21 $RAND_{20}$-$REG_1$, 21 $RAND_{20}$-$REG_2$, 21 $RAND_{20}$-$REG_5$, 21 $RAND_{20}$-$REG_{10}$, and 84 $RAND_{20}$, with one-third of the sequences containing a gap.

**Quantification and statistical analysis**. Comparison of PDRs across conditions: a series of paired t-tests were conducted on each pair of conditions (two-tailed; over the entire epoch length; downsampled to 20 Hz), with family-wise error (FWE) control using a non-parametric permutation procedure with 5000 iterations (cluster-defining height threshold of $p < 0.05$ with an FWE-corrected cluster size

threshold of $p < 0.05$[64]), as implemented in the Fieldtrip toolbox[65] (http://www.fieldtriptoolbox.org). Significant time intervals are presented as colored horizontal bars below the PDR plots.

Event rate analysis: because PD/PC events are rare (normal pre-transition rates are 1–2 per second) the statistical analysis was conducted by pooling over Experiment 1A and B (32 subjects overall). The cluster analysis used to compare PDR was conducted between each transition condition and its control with other details as described above.

Experiment 3: to quantify the change in PDR peak latency for STEP and $REG_{10}$-$RAND_{20}$ in Experiment 3A, B (active transition detection) relative to Experiments 1, 2, and 4 (gap detection), bootstrap analysis[66] (1000 iterations) was performed on two sets of participants, one constructed from the 14 active participants in Experiment 3 (active) and another from the 57 participants pooled from Experiments 1, 2, and 4 (non-active). On each iteration, a simulated PDR latency was computed over 14 participants randomly drawn from the non-active pool. The scatterplots in Fig. 6c show the distribution of the simulated peak latency of STEP (left) and $REG_{10}$-$RAND_{20}$ (right) in the non-active pool. The red and green crosses indicate the mean peak latency in Experiments 3A, B respectively.

Experiment 3A: we analyzed potential single-trial level associations between reaction time and pupil responses using REML in JMP 13.2 (SAS Institute, SAS Institute Inc., Cary, NC). Because reaction times (RTs) and pupillometry measures were non-normally distributed, rank scores were used for all analyses (ties assigned the bottom rank from the set of same values) with subjects as random effect and reaction time as fixed effect. Three measures of PDR dynamics were investigated: 'Max PDR amplitude', reflected the peak pupil diameter; 'Max PDR latency', reflected the latency of the PDR peak, 'max derivative PDR latency' measured the time of maximum rate of change. Participant was entered as the random factor, and rank RT as fixed effect; for reported analyses, slope was fixed over subjects to avoid potential overfitting, but all effects at $p < 0.05$ also hold when separate slopes are fit for each participant (except when noted). We report t- and p-values associated with the RT fixed effect, using the Satterwaithe approximation to estimate degrees of freedom (SAS Institute Inc 2017). We also provide an estimate of relative contribution of the fixed effect to overall model fit by computing partial $R2$ estimates using the lme4 and r2glmm[67] packages in R; these provide an implementation of the Nakagawa and Schielzeth algorithm[31]. Finally, we also verified the JMP-based estimates of the fixed effects using R packages lme4 and lmerTest[68].

**Reporting summary**. Further information on research design is available in the Nature Research Reporting Summary linked to this article.

## Data availability

Stimulus examples as well as all raw and processed data and the analysis code to reproduce the figures, are available at [https://doi.org/10.5522/04/c.4590887.v1]. A reporting summary for this Article is available as a Supplementary Information file.

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

## Acknowledgements

We are grateful to Alex Billig and Shihab Shamma for comments and discussion and to Makoto Yoneya for initiating the pupillometry setup at UCL. This work was supported by a BBSRC project grant and a BBSRC international partnering award to M.C.; S.Z. was partly funded by an NTT-UCL enhanced research contract.

## Author contributions

S.Z. conceived performed and analyzed the experiments; and wrote manuscript. M.C. supervised and administered the project; secured the funding; provided expertize; conceived the experiments and oversaw the analysis; and wrote the manuscript. F.D. conceived the experiments; formal analysis of Experiment 3A; and wrote the manuscript. P.D. provided expertize and feedback; and wrote the manuscript. S.F. provided expertize and feedback; and commented on manuscript draft. H.L. performed pilot experiments; provided expertize and feedback; commented on manuscript draft.

## Additional information

**Competing interests:** The authors declare no competing interests.

