## [Peer Review File · Nature Communications]

Reviewers' Comments:

Reviewer #1:

Remarks to the Author:

Review of "Phasic norepinephrine is a neural interrupt signal for unexpected events in rapidly unfolding sensory sequences – evidence from pupillometry"

The experiments reported in this paper used pupil dilation responses to sound sequences to probe LC-NE system responses to changes in statistics within rapid sensory signals. The paper yields a rich and informative set of findings regarding what triggers pupil dilation.

In particular, the authors compare regular-random and random-regular transition types. They argue that both transitions involve salient state changes but only the former is a punctate change. However, it seems to me that the random-regular change would be less salient. In perceptual research, salience can be defined as the amount of contrast between a stimulus and everything around it. It seems that the regular sequence should contrast less with the random sequence if it comes after it than vice versa. At the moment of the transition, the regular sequence fits fine with the expectation of something random. Then when it starts repeating, it still could fit (randomly, something could repeat); the regularity only becomes apparent over time which seems likely to diminish its salience. To make the argument that the two sequence transitions have the same salience, more discussion and a corresponding definition of salience are needed.

It was a nice finding that the random-regular transition does evoke a significant pupil dilation response when the transition is task relevant, even though it does not when it is not task relevant.

p. 23: Please clarify the direction of the association: "RT was significantly associated with maximum PDR amplitude ($t(334.9) = 3.06, p = 0.0024$)". Clarifying the direction of the relationship would be helpful in the next paragraph on the same page, as well.

Likewise in Figure 6, it is not specified what the ranking of RTs mean (i.e., are smaller values faster?)

p. 30: "such that it required double the power to reach significance" - doubling the N does not necessarily double the power so this statement is inaccurate.

p. 40; "Over time, and as the first author became more experienced with conducting the experiments, it was concluded that smaller N and fewer trials per condition were sufficient." - please report power analyses to indicate whether the N's are justified. Obtaining significant effects is not in itself a justification.

It was helpful as a reviewer to have the stimuli available. The authors state that they plan to share their data upon publication which is also a strength; hopefully they will include the stimuli in the public repository for other investigators to access.

Mara Mather

I sign all reviews

Reviewer #2:

Remarks to the Author:

Zhao et al examine hypothesis that the LC/NE system signals unexpected uncertainty through a

detailed set of pupillometry experiments. The authors take advantage of an interesting set of auditory stimuli that allow them to create rapid perceptual transitions that differ in the degree to which they abruptly violate expectations. They embed these stimuli in an alternate task to remove decision/response related confounds and find that only stimuli that abruptly violate concrete expectations give rise to transient pupil responses. The authors do a number of studies to verify that the differential pupil responses are not due to task performance differences, baseline pupil differences, or event timing differences.

Overall I found this paper to be interesting, well-written, and extremely thorough. While the authors are not the first to tackle the unexpected uncertainty hypothesis of NE function through pupillometry, they take an interesting and novel approach (dissociate expectancy violations from behavior, test asymmetry in expectation violations) and they provide a much more thorough characterization of the factors affecting transient pupil responses to expectancy violations than I have seen elsewhere. That said, there are a few concerns with the current manuscript that affect the degree to which the experimental results support the authors' interpretations of them and the degree to which the results inform our general understanding of the pupil-linked arousal system more generally.

1) I understand the point that the authors are making with regards to the asymmetry across RAND-REG/REG-RAND conditions, however in order for the results of these experiments to generalize to other experimental paradigms it would be useful if the authors could characterize this distinction in more concrete terms. For example, could the asymmetry across these stimulus conditions be quantified by running an inference model across the stimulus train and quantifying instantaneous Shannon information or KL divergence? I really appreciate that the authors have shown examples of each stimulus type (and made actual stimuli available online) however I think that some characterization of these stimuli in more information theoretic or probabilistic terms would be important to extrapolating the experimental findings from these experiments to other stimulus domains.

2) I am unconvinced by the claim that the results of experiment 3 are related to attention and not to the motor response itself. The authors show clearly that the pupil response does relate to the motor response and the interpretation of the fraction of variance explained by the motor response should include consideration of the overall correlation between the pupil and the LC/NE system, which on a trial to trial basis, is exceedingly small (Joshi & Gold 2016). If the authors wish to make a claim about attention to the transitions in the absence of a motor confound they should use a paradigm where the participant is required to make a post-listening indication of whether a transition occurred over an extended stimulus sequence (eg. Einhauser 2008).

3) The authors show that the baseline responses in the RAND condition are slightly elevated relative to the REG condition. The authors claim that the lack of statistical significance for this relationship in one experiment suggests that the baseline differences do not contribute to the transient response differences they measure, but I think this logic is a bit misleading for a few reasons: 1) measures of transient responses involve baseline subtraction, 2) the baseline diameter includes additional low frequency sources of variability that make detecting effects in baseline more difficult than detecting transient effects [where low frequency noise is removed], 3) and the more general statistical logic regarding interpreting non-significant results as different from significant ones. I think that this issue could be resolved by regressing out trial-wise baseline pupil diameter from transient responses rather than the current baseline subtraction method (Krishnamurthy 2017).

Minor comments:

Figure 4 spectrograms seem to be partially occluded by labels.

Lines 576-578:

But see also: Jepma 2016, Jepma 2018

Lines 653-654:

It seems worth pointing out the link between this idea and other recent pupillometry studies (eg. Urai, Krishnamurthy, de Gee).

Reviewer #3:

Remarks to the Author:

The authors investigated the relationship between pupil diameter, which they interpreted as a proxy for norepinephrine (NE) release in the brain, and the detection of changes to auditory stimuli. In a series of four experiments that are a clever extension of oddball tasks long used to study pupil/arousal responses, the authors found that pupil diameter increases when auditory stimuli switch from a structured (REG) to a random (RAND) sequence, both when the stimuli are behaviorally relevant or irrelevant. Conversely, pupil diameter increases for RAND-REG switches only when the stimuli are behaviorally relevant and not when they are irrelevant. The authors propose that these effects are due to "unexpected uncertainty", signaling situations where an internal predictive model is contradicted by the current evidence, and needs to be updated.

Overall the authors provide compelling evidence from well-formulated and well-controlled experiments that pupils dilate more in the REG-RAND versus the RAND-REG condition when the stimuli are not behaviorally relevant. Thus, although pupil diameter effects are often observed when stimuli are behaviorally relevant, the split result between behaviorally relevant and irrelevant stimulus switches provides a more rigorous examination of the conditions under which the arousal system signals certain kinds of abrupt statistical changes in the environment.

I also had several major concerns that hopefully the authors can address in a revision:

1. In general, the size of an evoked change in pupil diameter is strongly dependent on the baseline pupil diameter, but this effect was not considered sufficiently throughout the paper. The fact that there was not, on average, a statistically significant difference in pupil diameter in no-transition conditions (e.g., Figs. 2C,D, 4E, and 7B) does not necessarily imply that trial-by-trial relationships between baseline and evoked pupil diameter did not affect the evoked results. Particularly for results like for experiment 1B (Figs. 2B and D), it is imperative to show that the apparently larger RAND baseline in 2D is not the reason for the apparently smaller RAND-REG transition in 2B. The plots of the timing of occurrence of pupil events in Fig. 3 are nice and are consistent with the finding of no reliable RAND-RED pupil modulation, but it still would be useful to specifically account for trial-by-trial changes in baseline on the magnitude of the transition-triggered effects, for example by measuring evoked modulations in terms of residuals to the main relationship between baseline and evoked magnitude.

2. The interpretations in terms of unexpected and expected uncertainty are somewhat vague and could be clarified. For example, the lack of pupil effects in the behaviorally irrelevant RAND-REG condition is interpreted as reflecting "expected – rather than unexpected - uncertainty, leading to gradual rather than abrupt model change." This is an important enough point that it would be nice to have more quantitative and precise descriptions of how those quantities (expected and unexpected

uncertainty) relate to the kinds of statistical manipulations that were used for the stimuli in this study and could, in principle, be applied to such stimuli. For example, what is the boundary between a gradual versus abrupt change that would be hypothesized to determine whether or not LC/pupil responds? This point is somewhat addressed by experiment 4b, but there the different REG versions are interpreted only in terms of qualitative descriptions of the stimuli ("the most basic regular pattern"... "as the regularity becomes more complex; e.g., as we add more elements to the regular pattern") and not in terms of what makes the uncertainty associated with a particular transition expected or unexpected. Is it just the time course of changes in uncertainty that determines the difference? How does this definition depend on the complexity/randomness of the first stimulus, and more importantly the complexity of the model that the subject uses as a model of that stimulus (or alternatively, under what conditions is the stimulus too random to promote model building and therefore the ability to readily recognize unexpected uncertainty)?

3. On a related note, it wasn't clear to me how to think of the effects of task relevance in terms of the distinction between expected and unexpected uncertainty. Given the interpretation of the task-irrelevant pupil modulations in terms of that distinction, it seems like a straightforward interpretation of the effects of adding task relevance (e.g., RAND-REG evokes a pupil change under those conditions) is that task relevance somehow changes the definition of/boundary between those two forms of uncertainty. Is that what was meant by this quote: "a consequence of a behaviorally-driven emergence of a category boundary between REG and RAND, a richer representation of the statistics of the patterns before or after the transition, or a threshold change for model reset"? It would be useful to unpack that a bit. Or is the proposal that the principles that govern arousal responses are different for attended/task-relevant versus un-attended/task-irrelevant stimuli?

4. Another way to think about the differences between experiments 1/2 versus 3 is that in experiment 3, there is a much larger pupil diameter in the RAND versus other conditions when measured from stimulus onset. This effect may suggest something like increased uncertainty/more mental effort (both of which are known to modulate pupil diameter) from attentive subjects trying to identify structure in the RAND sequence. Might the fact that such modulation is not found in the task-irrelevant conditions imply that the brain simply is not putting in any effort to form a model of something that is apparently random?

A counter-argument to this idea is the MEG data from the previous study, showing that there are brain signals that represent the unattended RAND-REG transition (Fig. 1). Is it possible that the task differences (visual task in the MEG study, gap detection in this study) were a factor, such that in the current study the RAND stimuli were more likely to be ignored?

5. I am not sure what to take away from the data relating RT to pupil. There seem to be many conditions in which pupil was reliably modulated by RT. However, these results are downplayed ("these effects were not strongly linked to the execution of a motor command") and described as "modest... with RT accounting for between ~2-6% of estimated variance." Unfortunately, no reference is given for how much of the variance is accounted for by other factors, making it difficult to assess the motor component. Moreover, these effects are interpreted only in terms of the button press itself, as opposed to a more general role for stronger arousal modulations under behaviorally relevant conditions. It would be useful to interpret these findings in terms of previous findings that have related oddball-related changes in pupil diameter to RT on respond versus no-respond conditions (e.g., Kamp and Donchin, 2015, "ERP and pupil responses to deviance in an oddball paradigm").

6. As the authors are aware, there are several lines of evidence linking pupil diameter to LC activation, but the relationship is not necessarily specific (pupil also can reflect other neuromodulatory and neural

activation patterns) and has not been characterized under a broad range of conditions. Therefore, it would seem far more prudent for the title and language throughout to be more precise in describing the results in terms of modulations of arousal/pupil, and leave the putative link to the LC-NE system to the introduction and discussion.

Minor points/comments:

1. The word "punctate" is used throughout, apparently as a synonym for "sudden" or "abrupt", but to my knowledge means something closer to "related to perforations."
2. Line 183: the p-values for both CONST and REG10 are identical – is this correct?
3. The authors report p-values from linear mixed effects models (e.g., p23, line 418), but determining the exact degrees of freedoms for these models to compute t-values is challenging, and different methods exist with varying degrees of support (e.g., Kuznetsova, et al., 2018, J. Stat. Soft.). How were these p-values computed?
4. Unless I am mistaken in what is being plotted, it would be more clear if the y-axis in figures 2a-d and 4a-e were changed to "Pupil change from baseline" rather than "Pupil Diameter", given that these plots refer to baseline-corrected pupil changes (rather than their actual z-scored values).

Response to reviewers

We thank the reviewers for their careful reading and constructive feedback on our manuscript. We have considered all their comments and are grateful for the opportunity to resubmit a revised version of our manuscript. We address each of the reviewer points in turn, with details on the associated changes to the paper.

In particular, we have added a new experiment (Exp3B) to address Rev2 and Rev3's comments about understanding the relationship between the emergence of the PDR under active transition tracking conditions and the motor response. In this new experiment we demonstrate the presence of the PDR under delayed response conditions, confirming that it is not a consequence of the button press (see details below).

Reviewer #1

Q1: the authors compare regular-random and random-regular transition types. They argue that both transitions involve salient state changes but only the former is a punctate change. However, it seems to me that the random-regular change would be less salient. In perceptual research, salience can be defined as the amount of contrast between a stimulus and everything around it. It seems that the regular sequence should contrast less with the random sequence if it comes after it than vice versa. At the moment of the transition, the regular sequence fits fine with the expectation of something random. Then when it starts repeating, it still could fit (randomly, something could repeat); the regularity only becomes apparent over time which seems likely to diminish its salience. To make the argument that the two sequence transitions have the same salience, more discussion and a corresponding definition of salience are needed.

A1: Our original claim focused on the fact that the two transitions are detected around the same time both when measured behaviourally and by using MEG brain imaging in naïve distracted listeners. However, the reviewer is certainly correct that it is difficult to compare the salience of RAND-REG vs. REG-RAND transitions. Indeed if one takes the presence of a PDR to indicate increased arousal, then our main result – that RAND-REG transitions do not evoke a PDR in passive listening – could be taken to suggest that on some level, RAND-REG transitions are less salient than REG-RAND.

We have carefully re-read the text to ensure that we do not make any claims about equivalent salience. Thank you for pointing this out.

Q2: p. 23: Please clarify the direction of the association: “RT was significantly associated with maximum PDR amplitude ($t(334.9) = 3.06, p = 0.0024$)”. Clarifying the direction of the relationship would be helpful in the next paragraph on the same page, as well.

A2: The direction of the association is positive: longer RTs were associated with greater maximum PDR amplitudes. The text has been amended to clarify this point.

Q3: *Likewise in Figure 6, it is not specified what the ranking of RTs mean (i.e., are smaller values faster?)*

A3: Yes, the smaller rank values mean smaller RT values. The figure legend has been amended to clarify this point.

Q4: *p. 30: "such that it required double the power to reach significance" - doubling the N does not : necessarily double the power so this statement is inaccurate.*

A4: The reviewer is entirely correct. We have changed the text to remove this statement, and thanks for flagging this.

Q5: *p. 40; "Over time, and as the first author became more experienced with conducting the experiments, it was concluded that smaller N and fewer trials per condition were sufficient." - please report power analyses to indicate whether the N's are justified. Obtaining significant effects is not in itself a justification.*

A5: The following text was added to the methods section (page 47):

"We ran a power analysis on Expt1A and Exp1B which were chronologically the first and second experiments in this project. The power analysis was conducted in the G*Power software package (Faul et al. 2009), with the following settings: $1 - \beta > 0.8$, and $p = 0.05$. To produce a measure of the magnitude of the PDR effect, we first found the peak PDR latency in the grand average for STEP and REG-RAND respectively (Figure 2A, B), and used this value to obtain, for each subject, the amplitude in the transition conditions and their respective controls. A 'net effect' measure (quantifying the size of the PDR) was then computed for each subject by taking the difference between each transition and its control (no-change) condition. Note that this manner of reducing the PDR effect to a single number per subject is necessarily much more conservative than the time-sensitive analysis employed to quantify the PDR in the main analysis.

Both experiments enjoyed large effects sizes: Cohen's $d \sim 1.2-1.7$ for STEP and $d=0.7-0.8$ for RAND-REG. Using an estimated effect size of 0.8 yielded an $N=12$, confirming all experiments are adequately powered."

Q6: *It was helpful as a reviewer to have the stimuli available. The authors state that they plan to share their data upon publication which is also a strength; hopefully they will include the stimuli in the public repository for other investigators to access.*

A6: Yes, indeed. We will share all the materials and data that we collected.

Reviewer #2

Q1: *1) I understand the point that the authors are making with regards to the asymmetry across RAND-REG/REG-RAND conditions, however in order for the results of these experiments to generalize to other experimental paradigms it would be useful if the authors could characterize this distinction in more concrete terms. For example, could the asymmetry across these stimulus conditions be quantified by running an inference model across the stimulus train and quantifying instantaneous Shannon information or KL divergence? I really appreciate that the*

authors have shown examples of each stimulus type (and made actual stimuli available online) however I think that some characterization of these stimuli in more information theoretic or probabilistic terms would be important to extrapolating the experimental findings from these experiments to other stimulus domains.

A1: Thank you for the suggestion, which we have implemented, and show in the figures below (also added as ‘Supplementary Figures S1 and S2’ to the manuscript).

To quantify the predictability of each tone-pip within the sequences, we applied a model of auditory expectancy (Information Dynamics Of Music; IDyOM; Pearce, 2005) as a theoretical benchmark. The model is based on multiple-viewpoint, variable-order Markov chains; for each tone in a sequence, it outputs information content (IC) as a measure of unexpectedness, given the preceding context. This model is sensitive to sequential regularities and is hence a suitable model for quantifying the statistics of the present stimuli. It was also previously used in the context of an MEG study based on the same stimuli (Barascud et al. 2016).

We applied IDyOM to the stimuli in Experiment 1A (Figure R1) and Experiment 4B (Figure R2). For each, the model was ran on the entire experimental session, with stimuli in different conditions presented in a random order (in the same way they were delivered to the human participants). We used the LTM+ model configuration, which initiates with an empty model, then learns over the stimulus set, updating the model after each tone. This mirrors the experience of the human participants who likely learned the probability structure of the stimuli throughout the course of the experimental session.

Plotted in Figures R1 and R2 is the information content associated with the last trial for each of the stimulus categories. The output (shown in coloured lines, with shading = 2SD) thus reflects the probability associated with each tone in light of the most ‘complete’ internal model. The dashed black line plots the Bhattacharyya coefficient (BC) between each pair of conditions. The BC is a measure of the degree of overlap between the distribution of IC, such that BC=1 indicates perfect overlap BC=0 indicates zero overlap.

As can be seen from the figures, the transition in STEP and REG₁₀-RAND₂₀ is associated with an immediate change in IC. In contrast, transitions from RAND are associated with a gradual decrease in IC over several tones.

Figure R1. Modelling results for the stimulus set in Experiment 1A. STEP, REG₁₀-RAND₂₀ and RAND₂₀-REG₁₀ and their controls. The information content for each tone from the chosen single trial (always the last presentation of each condition within the

experimental session) is shown against the tone number relative to transition. To focus on the transition period, we plot the interval from 5 tones before the transition to 15 tones after the transition. As in Figure 1 in the main text, in $RAND_{20}$ - REG_{10} sequences, the transition time is defined as occurring after the first full regular cycle, i.e. once the transition becomes theoretically detectable. Shading indicates 2 standard deviations from the average across all trials. The black dashed curve indicates the Bhattacharyya coefficient—an estimate of the amount of overlap between the two distributions—labelled on the right y-axis. For STEP and REG_{10} - $RAND_{20}$ the ideal observer model shows an effectively instantaneous detection of the transition. For in $RAND_{20}$ - REG_{10} , this ideal observer model requires about 4-5 tones (after the 'effective transition') to discover the emergence of regularity. Intriguingly, active listeners also required a similar number, 277ms (5.5 tones), to detect the transition $RAND_{20}$ - REG_{10} , suggesting that human performance is comparable with an ideal observer gifted with perfect memory and processing resources.

Figure R2. Modelling results for the stimulus set in Experiment 4B: $RAND_{20}-REG_1$, $RAND_{20}-REG_2$, $RAND_{20}-REG_5$, $RAND_{20}-REG_{10}$ and their control $RAND_{20}$. The information content for each tone from the chosen single trial (always the last presentation of each condition within the experimental session) is shown against the tone number relative to transition. To focus on the transition period, we plot the interval from 5 tones before the transition to 15 tones after the transition. As in Figure 1, in the main text, the transition time is defined as occurring after the first full regularity cycle. Shading indicates 2 standard deviations from the average across all trials. The black dashed curve indicates the Bhattacharyya coefficient—an estimate of the amount of overlap between the two distributions—labelled on the right y-axis. All transition conditions show a gradual decline in IC over several tones, though the drop in $RAND_{20}-REG_1$ appears to occur one tone earlier, on average. The temporary dip in IC around tone#9 in the $RAND_{20}$ condition reflects an idiosyncrasy of the single trial plotted (note that the same trial is replotted in all sub-panels because it was the last $RAND_{20}$ trial presented) and reflects the fact that the sequence happened to contain an arrangement of tones that occurred previously in another trial and was therefore ‘familiar’ to the model.

Q2: 2) *I am unconvinced by the claim that the results of experiment 3 are related to attention and not to the motor response itself. The authors show clearly that the pupil response does relate to the motor response and the interpretation of the fraction of variance explained by the motor response should include consideration of the overall correlation between the pupil and the LC/NE system, which on a trial to trial basis, is exceedingly small (Joshi & Gold 2016). If the authors wish to make a claim about attention to the transitions in the absence of a motor confound they should use a paradigm where the participant is required to make a post-listening indication of whether a transition occurred over an extended stimulus sequence (eg. Einhauser 2008).*

A2: We ran another experiment (N=14), similar to the one suggested by the reviewer, which has now been added to the manuscript as Experiment 3B. In this version of the task, participants were instructed to monitor sequences for pattern changes and to indicate their response at the end of the trial.

Below is the newly added text (p 26) and the associated results figure. We’ve also edited the methods section (p) giving the details of the new experiment.

“To confirm that the PDR to $RAND_{20}-REG_{10}$ observed in Exp3A is indeed not confounded by the motor response, we repeated the experiment using a delayed response paradigm: Participants were instructed to monitor the tone sequences for pattern changes but indicate their response at the end of the trial. To control for vigilance and discourage participants from only attending to the beginning and end of a sequence, they were also instructed to monitor the stimuli for silent gaps, which could occur at any time (as in Exp1,2,4; see methods). The subset of sequences containing gaps or any button presses were excluded from the pupillometry analysis.

Behavioral results are summarized in Fig. 7A,B. For the change detection task (Fig 7A), hit and false alarm rate data demonstrated that performance on all transition conditions

was at ceiling, with no difference between conditions ($F(1.99,25.9)=1.62$ $p=0.216$). The false alarm effects observed in Exp3A were not seen here, likely because of the delayed response nature of the task. The gap detection data (Fig 7B) also revealed no difference across conditions ($F(1.91,24.8)=2.86$ $p=0.078$).

The Pupillometry analysis results were likewise generally consistent with those observed in Exp3A: Fig 7C plots the average normalized pupil diameter relative to sequence onset. In line with Exp3A, and in contrast to Exp1 and 2, the sustained pupil diameter to RAND₂₀ rose above that to REG₁₀, likely reflecting the increased perceptual or computational demands associated with tracking random, relative to regular, patterns (see discussion). However, these effects were somewhat more modest overall, including no difference between REG₁₀ and CONST, despite that seen in Exp3A. These moderate effects likely relate to the more relaxed tracking nature of the delayed response task, which may not have required the same level of vigilance and/or close tracking of sequence structure as that in Exp3A.

The PDRs, shown in Fig 7D, were also reduced relative to those observed in Exp3A. Importantly, however, clear pupil dilations were observed in all three change conditions, including RAND₂₀-REG₁₀. Together with the results from Exp3A this confirms that the emergence of the PDR to RAND₂₀-REG₁₀ is not a consequence of the motor response (or the decision to act, assuming the two are highly correlated³⁰). Rather, having listeners actively monitor and respond to the statistical transitions prompted a change in the underlying cognitive process, e.g. by delineating the category (or decision) boundary between RAND and REG, and thereby rendering the transition as a model violation. We return to this point in the discussion. “

Figure R3. Experiment 3B (N=14): Active transition detection (delayed response). [A] Transition detection task: Hit rates and false positive rates. Circles indicate individual subject data; error bars are ± 1 SEM. [B] Gap detection task. Circles indicate individual subject data; error bars are ± 1 SEM. [C] Average pupil diameter relative to sequence onset. RAND₂₀ statistically diverged from CONST at 1260ms and from REG₁₀ from 2520ms post-onset. [D] Average pupil diameter relative to the transition. Solid lines represent the average normalized pupil diameter, relative to the transition. Shading shows ± 1 SEM. Coloured horizontal lines indicate time intervals where cluster-level statistics showed significant differences between each change condition and its control. The PDR to STEP increased from ~ 200 ms post-transition, peaking at 1520ms; it statistically diverged from CONST between 900ms and 2360ms. For REG₁₀-RAND₂₀, the responses commenced ~ 500 ms post-transition, peaking at 1800ms, and statistically diverged from REG at 740ms post-transition through to sequence offset. For RAND₂₀-REG₁₀, the response emerged at

820ms, peaked at 2020ms, and statistically diverged from its control $RAND_{20}$ between 1500 and 2320ms. Comparing transition conditions directly, no difference was observed between STEP and $REG_{10}-RAND_{20}$, $REG_{10}-RAND_{20}$ was significantly greater than $RAND_{20}-REG_{10}$ between 1240ms and 2060ms post-transition.

Q3: 3) *The authors show that the baseline responses in the RAND condition are slightly elevated relative to the REG condition. The authors claim that the lack of statistical significance for this relationship in one experiment suggests that the baseline differences do not contribute to the transient response differences they measure,*

A3: We believe the reviewer might have misunderstood this aspect of the results, and apologise that this might not have been clear. We found no statistically significant difference between RAND and REG in any of the FOUR gap detection experiments (Experiments 1A, 1B, 2 and 4). Significant differences between RAND and REG were only seen in the two active transition detection experiments (Exp3A and the newly added Exp3B). We now make this clear in the paper.

Q4: *but I think this logic is a bit misleading for a few reasons: 1) measures of transient responses involve baseline subtraction, 2) the baseline diameter includes additional low frequency sources of variability that make detecting effects in baseline more difficult than detecting transient effects [where low frequency noise is removed], 3) and the more general statistical logic regarding interpreting non-significant results as different from significant ones. I think that this issue could be resolved by regressing out trial-wise baseline pupil diameter from transient responses rather than the current baseline subtraction method (Krishnamurthy 2017).*

A4: Thank you for these suggestions. Firstly, we note that an analysis akin to the one in Krishnamurthy et al (2017) is not applicable to the present paradigm. In their study, they regressed out the baseline diameter from the correlation between the PDR and ‘surprise’. Here, ‘surprise’ is not parametrically modulated (rather we have 3 different transition conditions). Instead, our approach has been to compare transition conditions with their no-change controls.

Instead of quantifying the pupil diameter by a single measure (e.g. max pupil diameter), we analysed the raw PDR data by comparing transition vs no transition conditions. Therefore, baseline correction (within participant) was necessary to quantify the PDR. The conclusions are focused on the presence vs. absence of the PDR at the subject/group level and demonstrate the presence of a PDR for RAND-REG (and STEP) conditions but a consistent lack of PDR for the opposite condition.

The evidence for the fact that baseline difference is not the underlying source of the lack of a PDR to RAND-REG in the gap detection experiments (Exp 1,2,4) is:

(1) The consistent lack of significant baseline differences between RAND and REG observed in Experiment 1,2,4.

(2) The fact that a PDR to RAND-REG was present in Experiment 3 (active transition detection) despite a large pre-transition difference between RAND and REG in that experiment.

However, to address the reviewer’s request, we computed a correlation between the pre-transition amplitude and PDR magnitude on a single trial basis (Experiment 1). The raw

pupil diameter data were first baseline corrected over the 1s pre-sequence onset. Epochs around the transition (from one second before to 2 seconds after the transition) were extracted and z-score normalised across all conditions within each block in each subject. For each trial, the average pupil diameter was then computed over two time windows:

W1: During the pre-transition interval (1 second)

W2: 0.5s to 1.5s after the transition (the interval during which the PDR is observed).

Trials with outlier pre-transition amplitudes (2 standard deviations away from the condition mean) were removed. Trials were then pooled across all 18 subjects. We correlated W1 with (W2-W1; as a measure of the PDR).

Figure R4 presents the results of this analysis. As expected, the baseline amplitude is significantly (but rather weakly) negatively correlated with the PDR in STEP ($r=-0.205, p<0.0001$) and REG-RAND ($r=-0.173, p=0.003$), but not in RAND-REG and other no-change conditions ($p>0.1$). This single-trial based result is consistent with the group average result reported in the main text: i.e. the presence of a PDR in STEP and REG-RAND but not in RAND-REG.

Figure R4. Single-trial relationship between pre transition amplitude and the PDR. Scatter plots show the baseline amplitude (z-score normalised) for each trial (y-axis) versus the average post-transition change in amplitude for the same trial (x-axis), separated by condition in Expt1A. The associated Spearman's r and p values are given above each scatterplot.

With respect to establishing the null effect for RAND-REG: We take the point that interpreting non-significant results as different from significant results is problematic. We took the approach of many replications to strengthen the argument of a lack of PDR to RAND-REG. If the reviewer feels it is useful, we can also add the following analysis to the results section of Experiment 1:

“A repeated measures ANOVA over the instantaneous PDR net effect (difference between each transition stimulus and its control at the transition’s group peak PDR latency; N=32 collapsed across data from Exp1A,B) showed a main effect of condition ($F(2,62)=20.5$, $p<0.0001$). Post hoc, Pairwise comparisons demonstrated that the net effect was different across each pairs of conditions ($p<0.002$) such that $STEP>REG-RAND>RAND-REG$. A one-sample t-test (2-tailed) showed that the PDR net effects of STEP and REG-RAND were significantly different from 0 ($t=7.4$, $t=4.3$ respectively, $p<0.0001$), whilst that for RANDREG was not ($t=0.6$ $p=0.53$)”

Q5: Figure 4 spectrograms seem to be partially occluded by labels.

A5: changed

Q6: Lines 576-578: But see also: Jepma 2016, Jepma 2018

A6: References included.

Q7: Lines 653-654: It seems worth pointing out the link between this idea and other recent pupillometry studies (eg. Urai, Krishnamurthy, de Gee).

A7: Thank you for this suggestion. We have added a references to de Gee et al.

Since the statement which the reviewer indicates actually refers to brain imaging data we felt the other two references may not be directly pertinent. Krishnamurthy is already referenced in several places in the manuscript. We have also added a reference to Urai et al (page4).

Reviewer #3

Q1: 1. In general, the size of an evoked change in pupil diameter is strongly dependent on the baseline pupil diameter, but this effect was not considered sufficiently throughout the paper. The fact that there was not, on average, a statistically significant difference in pupil diameter in no-transition conditions (e.g., Figs. 2C,D, 4E, and 7B) does not necessarily imply that trial-by-trial relationships between baseline and evoked pupil diameter did not affect the evoked results. Particularly for results like for experiment 1B (Figs. 2B and D), it is imperative to show that the apparently larger RAND baseline in 2D is not the reason for the apparently smaller RAND-REG transition in 2B. The plots of the timing of occurrence of pupil events in Fig. 3 are nice and are consistent with the finding of no reliable RAND-RED pupil modulation, but it still would be useful to specifically account for trial-by-trial changes in baseline on the magnitude of the transition-triggered effects, for example by measuring evoked modulations in terms of residuals to the main relationship between baseline and evoked magnitude.

A1: Please see our response to a similar comment by reviewer 2, Above (Q4).

Q2: 2. *The interpretations in terms of unexpected and expected uncertainty are somewhat vague and could be clarified. For example, the lack of pupil effects in the behaviorally irrelevant RAND-REG condition is interpreted as reflecting “expected – rather than unexpected - uncertainty, leading to gradual rather than abrupt model change.” This is an important enough point that it would be nice to have more quantitative and precise descriptions of how those quantities (expected and unexpected uncertainty) relate to the kinds of statistical manipulations that were used for the stimuli in this study and could, in principle, be applied to such stimuli. For example, what is the boundary between a gradual versus abrupt change that would be hypothesized to determine whether or not LC/pupil responds? This point is somewhat addressed by experiment 4b, but there the different REG versions are interpreted only in terms of qualitative descriptions of the stimuli (“the most basic regular pattern”... “as the regularity becomes more complex; e.g., as we add more elements to the regular pattern”) and not in terms of what makes the uncertainty associated with a particular transition expected or unexpected. Is it just the time course of changes in uncertainty that determines the difference? How does this definition depend on the complexity/randomness of the first stimulus, and more importantly the complexity of the model that the subject uses as a model of that stimulus (or alternatively, under what conditions is the stimulus too random to promote model building and therefore the ability to readily recognize unexpected uncertainty)?*

A2: We agree with the reviewer that these are key questions. We hope that the IC modelling which we now provide (see Reviewer 2, Q1) will address some of these issues.

Answers to the deeper questions, related to characterizing the boundary between ‘expected’ and ‘unexpected’ uncertainty are beyond the scope of the present paper. We hope that our demonstration of the presence of such a boundary will encourage others to pursue these issues. Some relevant work is planned in our lab (PI: Maria Chait) where we will systematically manipulate the statistics of the pre-transition signals and measure pupil, behavioural, and brain responses.

We are careful to word the conclusions of this study to highlight these outstanding issues. This text (page 36) now reads:

“The demonstration of a distinct boundary between ‘expected’ and ‘unexpected’ uncertainty calls for future modeling and experimental work to outline the properties of this distinction and its implications for perception.”

Q3: 3. *On a related note, it wasn’t clear to me how to think of the effects of task relevance in terms of the distinction between expected and unexpected uncertainty. Given the interpretation of the task-irrelevant pupil modulations in terms of that distinction, it seems like a straightforward interpretation of the effects of adding task relevance (e.g., RAND-REG evokes a pupil change under those conditions) is that task relevance somehow changes the definition of/boundary between those two forms of uncertainty. Is that what was meant by this quote: “a consequence of a behaviorally-driven emergence of a category boundary between REG and RAND, a richer representation of the statistics of the patterns before or after the transition, or a threshold change for model reset”? It would be useful to unpack that a bit. Or is the proposal that the principles that govern arousal responses are different for attended/task-relevant versus unattended/task-irrelevant stimuli?*

A3: We have rewritten the relevant text to unpack the argument. In short, both of the interpretations proposed by the reviewer are possible. Future work using a combination of brain imaging, pupillometry and behavioural task manipulation may be able to dissociate between them.

The amended text in page 38 now reads:

“Rendering the sequence transitions behaviorally relevant resulted in marked differences in pupil response dynamics. Most notably, active monitoring gave rise to a PDR to RAND₂₀-REG₁₀ transitions. These effects were not strongly linked to the execution of a motor command, as evidenced by the fact that RT accounted for relatively little variance in various PDR metrics and that the effects were largely preserved in a delayed response version of the task. Therefore, these behavior-related changes in the PDR likely reflect a change in the functional state of the LC-NE system, or of inputs to it. For example, it is possible that task relevance, or heightened arousal under behaviourally relevant conditions, leads to a richer representation of the statistics of the RAND patterns or contributes to the emergence of a category boundary between REG and RAND patterns thereby rendering the transitions, in both directions, as model violations. Alternatively, behavioral relevance may alter the boundary between “expected” and “unexpected” uncertainty, resulting in a threshold change for model reset. “

Q4: 4. Another way to think about the differences between experiments 1/2 versus 3 is that in experiment 3, there is a much larger pupil diameter in the RAND versus other conditions when measured from stimulus onset. This effect may suggest something like increased uncertainty/more mental effort (both of which are known to modulate pupil diameter) from attentive subjects trying to identify structure in the RAND sequence. Might the fact that such modulation is not found in the task-irrelevant conditions imply that the brain simply is not putting in any effort to form a model of something that is apparently random?

A4: Yes, this may be a possible interpretation as we note in the discussion section.

Page 38 (bottom) reads:

“Previous work has linked tonic pupil diameter differences to representation of expected uncertainty^{2,11} possibly driven by cholinergic signaling^{15,27}. The present effects may be consistent with this interpretation; as indeed RAND₂₀ is associated with less reliable priors than REG₁₀. However, the fact that these differences in pupil diameter were observed exclusively during the active change detection task must therefore suggest that cholinergic activation is dependent on behavioral relevance and is not involved in automatic tracking of sequence predictability. An alternative, but not mutually exclusive, possibility is that this effect may reflect heightened vigilance or listening effort⁶¹ arising through active sequence structure scanning, which is more demanding for RAND₂₀²⁵. “

We also note that the fact that a large PDR is observed in the RAND-REG condition, despite a remarkable difference in baseline between conditions (Figure 5), suggests that

baseline differences per se are not a likely explanation for the lack of a PDR in Experiments 1, 2 (Figures 2,4).

Q5: A counter-argument to this idea is the MEG data from the previous study, showing that there are brain signals that represent the unattended RAND-REG transition (Fig. 1). Is it possible that the task differences (visual task in the MEG study, gap detection in this study) were a factor, such that in the current study the RAND stimuli were more likely to be ignored?

A5: We believe that the task difference is not a likely explanation for the difference between the MEG and pupil data. We have replicated the MEG results with a range of tasks including passive listening (data currently prepared for publication), watching a silent movie (Southwell et al, 2017; 2018), and detecting occasional noise bursts between successive stimuli (Sohoglu & Chait, 2016).

As we note in the discussion session, we believe the results point to a disconnect between brain responses (which reflect active tracking of stimulus statistics) and pupil responses (reflecting the arousal system).

Q6: 5. *I am not sure what to take away from the data relating RT to pupil. There seem to be many conditions in which pupil was reliably modulated by RT. However, these results are downplayed (“these effects were not strongly linked to the execution of a motor command”) and described as “modest... with RT accounting for between ~2-6% of estimated variance.” Unfortunately, no reference is given for how much of the variance is accounted for by other factors, making it difficult to assess the motor component. Moreover, these effects are interpreted only in terms of the button press itself, as opposed to a more general role for stronger arousal modulations under behaviorally relevant conditions. It would be useful to interpret these findings in terms of previous findings that have related oddball-related changes in pupil diameter to RT on respond versus no-respond conditions (e.g., Kamp and Donchin, 2015, “ERP and pupil responses to deviance in an oddball paradigm”).*

A6: We believe that the Kamp & Donchin (2015) paper used a paradigm where a response was required on each trial. Perhaps the reviewer had another paper in mind?

The analysis focused on RT as this was the only available (single trial-level) factor.

The issue of overall arousal is a possibility and related to this reviewer’s comment #4, above. We have amended the text to include this point as detailed in our response to that comment.

Please also see the new Exp3B (see response to Reviewer 2, Q2), where we modified the behavioural paradigm to de-couple the motor response and transition detection.

Q7: 6. *As the authors are aware, there are several lines of evidence linking pupil diameter to LC activation, but the relationship is not necessarily specific (pupil also can reflect other neuromodulatory and neural activation patterns) and has not been characterized under a broad range of conditions. Therefore, it would seem far more prudent for the title and language throughout to be more precise in describing the results in terms of modulations of arousal/pupil, and leave the putative link to the LC-NE system to the introduction and discussion.*

A7: There is accumulating evidence (Joshi et al, 2016; Bitsios et al, 1996) to suggest that phasic pupil dilation is linked to LC activation, and since our hypotheses and data are

mostly focused on analysing phasic responses we feel a specific discussion in terms of the LC-NE system is warranted. That said, we have been careful to restrict references to LC-NE to the intro/discussion sections.

To address the reviewer's concern we have changed the title to:

“Pupil-linked phasic arousal evoked by violation but not emergence of regularity within rapid sound sequences”

Q8: 1. The word “punctate” is used throughout, apparently as a synonym for “sudden” or “abrupt”, but to my knowledge means something closer to “related to perforations.”

A8: All instances of ‘punctate’ have been replaced with ‘abrupt’

Q9: 2. Line 183: the p-values for both CONST and REG10 are identical – is this correct?

A9: Yes, this is correct, thanks for the close read.

Q10: 3. The authors report p-values from linear mixed effects models (e.g., p23, line 418), but determining the exact degrees of freedoms for these models to compute t-values is challenging, and different methods exist with varying degrees of support (e.g., Kuznetsova, et al., 2018, J. Stat. Soft.). How were these p-values computed?

A10: We provide a detailed description of the methods in the methods section. We note that the mixed model results are extremely similar when computed using SAS and R approaches, thus lending further support to the stability of these effects.

Q11: 4. Unless I am mistaken in what is being plotted, it would be more clear if the y-axis in figures 2a-d and 4a-e were changed to “Pupil change from baseline” rather than “Pupil Diameter”, given that these plots refer to baseline-corrected pupil changes (rather than their actual z-scored values).

A11: changed

Reviewers' Comments:

Reviewer #2:

Remarks to the Author:

Zhao and colleagues have revised their paper substantially and it is much improved. In particular, the new experiment completely addresses my concern about motor effect. I have two remaining concerns... both related to concerns that I raised about the previous manuscript.

1) The authors have characterized the moment-by-moment surprise in a markovian ideal observer across their stimuli, which clarifies some of the more general statistical differences between the conditions. However, I still feel that the bridge between the findings presented here, and more general real world stimuli is not made clear in the manuscript. Given the surprise profiles for each of the conditions, and the degree to which these conditions evoke pupil responses, is there something that could be said for about the degree to which a new stimulus sequence might generate a pupil response when presented either in a task context or outside of one? The authors have done a tremendous amount of work to characterize pupil responses to their own stimulus set – however, for this work to be of interest to a wider audience it is critical that the results from these studies make predictions that extend beyond the particular stimuli explored here.

2) I understand that the authors have seen 4 null results regarding baseline effects, but as I said in my previous review, null effects should not be interpreted as evidence for the null, but failure to reject it. One reason that the null might not be rejected is due to lack of power. One reason that the baseline analyses provided in the current manuscript may lack power is that they include low frequency variability in the pupil response that is unrelated to the underlying biological processes of interest (eg. Reimer 2015).

I don't understand the authors' response to my concern regarding baseline subtraction. There is no reason that the regression approach cannot be taken on categorical data. I am not saying that the authors need to change analyses throughout the paper, but given that baseline subtracted results can pick up on either side of the equation, it would be nice to clearly demonstrate that the effects that they are interested in emerge only from the positive side of the subtraction. As I said before, this could be done with a regression, or it could be done by showing that the category differences hold in PDR after regressing out pre-transition baseline from their PDR measure. Based on the data in R4, it seems unlikely that removing variance related to pre-transition baseline in the PDR will substantially change the results, but given that doing so is critical for interpretation, and that figure R4 is not included in the paper, it seems critical that the authors make the appropriate statistical argument to show that their effects are not driven by baseline differences.

Reviewer #3:

Remarks to the Author:

I would like to thank the reviewers for taking the time to address my concerns. Their additional experiment does indeed remove some of the concerns regarding the motor influence on pupil. With that said I still have a few concerns/comments, that I anticipate the authors will be able to address with a few additional analyses:

1) The authors may have misunderstood the comments reviewer 2 and I raised regarding baseline pupil. We highlight that PDRs are inherently dependent on the baseline size of the pupil on each trial. For example, PDRs will be smaller if baseline pupil is larger, simply by virtue of the fact that the PDR has less room to increase when the pupil is already dilated. To account for the influence of baseline

pupil on the PDRs, the authors should include baseline pupil (the actual baseline value) as a nuisance regressor in their PDR pupil analysis (similar to Krishnamurthy et al., 2017, Nat. Hum. Behavior) – i.e., in predicting the pupil change from baseline at each discrete timepoint in their PDR analyses, the authors could include a nuisance regressor of the value of the baseline pupil diameter.

2) I still think more should be discussed regarding whether the effects the authors observe, particularly the difference between Exp. 1 & 2 and Exp 3, can be broadly characterized as unexpected uncertainty. As I've stated in my previous comments, Exp 1 & 2 do not require some internal model of the stimuli, suggesting to me that the effects the authors observe are linked more to sensory effects (e.g., some form of repetition suppression) rather than violations of top-down expectations. When top-down expectations become a part of the task (e.g., Exp 3), the asymmetry between RAND-REG and REG-RAND disappears, providing stronger evidence for the role of unexpected uncertainty. I do find this distinction interesting and important, and feel like it should get more attention in the authors' interpretations of their results.

3) Regarding the p values for the linear mixed effects models, the authors indicate in the methods that these models were run in R using the lmer package, which does not, by default, provide a p-value. How were the p-values computed? I ask because p-values can be computed using likelihood ratios, but these have a tendency to increase type 1 error. The preferred method for computing p-values for LMEs is by using approximations such as the Satterthwaite method for computing degrees of freedom (see Kuznetsova, et al., 2018, J. Stat. Soft. – lmerTest package).

Response to reviewers

We thank the reviewers and the editor for their time and effort in careful reading and constructive feedback on our manuscript. Below, we address each of the reviewer points in turn, with details on the associated changes to the paper. In the new manuscript file, major changes to the text are indicated with grey shading.

In particular, with apologies for the previous misunderstanding, we have now re-analyzed all the data using the regression approach suggested by Rev2 and Rev3. All data figures and associated results have been updated. Whilst the new analysis led to small changes to various numerical values, it did not change any of the experimental outcomes. We hope that this new analysis provides a resolution to the issues of possible baseline noise.

Editor

Q1: Recall also that Reviewer 1 had suggested better justifying your sample. While we appreciate your addition of the power analyses on Experiments 1a and 1b, it is not clear to the editorial team whether the power analyses were conducted before or after collection of those data. We'd also like to note that the statement "it was determined that smaller N and fewer trials per condition were sufficient" is not clear. It would be helpful to explain on what basis this was determined.

A1: This is the first pupillometry experiment conducted on this type of acoustic stimulus. In the course of understanding the effect size it was important to determine the number of participants required, but also how many trials per subject are sufficient to yield a robust effect. This is a delicate balance because the quality of the data (a clean pupil signal, devoid of blink and other artefacts) inversely depends on the length of the experimental session and other related factors. There have been small adjustments (detailed in the methods section) made throughout the project to maximize data quality.

We also note that all of our experiments had an "in-built" self-replication such that the key findings are replicated over multiple experiments, to convince ourselves and the reader of their robustness.

We apologize that our statement about power analysis was not clear. This was indeed formally conducted following the request of reviewer #1. The power analysis was conducted based on effect size estimated from Experiments 1a and 1b, which were the first experiments conducted in this series. Both experiments enjoyed large effect sizes: Cohen's $d \sim 1.2-1.7$ for STEP and $d=0.7-0.8$ for RAND-REG. Using an estimated effect size of 0.8 for the power analysis yielded an $N=12$. This therefore confirmed that the number of participants in all subsequent experiments ($N \geq 12$ for all) would indeed have been sufficient.

To clarify all these issues text ('methods' section p 49) has been modified to read:

The experiments were not conducted in the order in which they are reported. Initial experiments involved larger numbers. Over time, and as we became more familiar with the variability across/between subjects, it was determined that smaller N and fewer trials per condition were sufficient to yield stable effects (see details for each experiment below). We

also note that key findings are replicated across multiple experiments, providing further evidence for effect robustness.

As a post-hoc confirmation of the adequacy of sample-sizes, we quantified effect sizes in Expt1A and Expt1B, which were chronologically the first and second experiments in this project, and used those values for a power analysis to derive N. The power analysis was conducted in the G*Power software package⁶¹, with the following settings: $1 - \beta > 0.8$, and $p = 0.05$.

To produce a measure of the magnitude of the PDR effect in Expt 1A, B, we first found the latency of the PDR peak in the grand average for STEP and REG-RAND respectively (Figure 2A, B), and used this value to obtain, for each subject, the amplitude in the transition conditions and their respective controls. A 'net effect' measure (quantifying the size of the PDR) was then computed for each subject by taking the difference between each transition and its control (no-change) condition. Note that this manner of reducing the PDR effect to a single number per subject is necessarily more conservative than the time-sensitive analysis employed to quantify the PDR in the main analysis.

Both Expt1A and Expt1B enjoyed large effects sizes: Cohen's $d \sim 1.2-1.7$ for STEP and $d = 0.7-0.8$ for RAND-REG. An overall estimated effect size of 0.8 was therefore fed into the power analysis and yielded an $N=12$, confirming that all further experiments ($N > 12$ for all) were adequately powered. "

Reviewer #2

Q1: The authors have characterized the moment-by-moment surprise in a markovian ideal observer across their stimuli, which clarifies some of the more general statistical differences between the conditions. However, I still feel that the bridge between the findings presented here, and more general real world stimuli is not made clear in the manuscript. Given the surprise profiles for each of the conditions, and the degree to which these conditions evoke pupil responses, is there something that could be said for about the degree to which a new stimulus sequence might generate a pupil response when presented either in a task context or outside of one? The authors have done a tremendous amount of work to characterize pupil responses to their own stimulus set – however, for this work to be of interest to a wider audience it is critical that the results from these studies make predictions that extend beyond the particular stimuli explored here.

A1: Thank you for raising this issue, we have now re-organized the initial section of the discussion to emphasize the main results and their broader implications. This now reads (p 34):

"A large body of work has suggested a gating role for NE in balancing bottom-up-driven sensory processing vs. top-down priors¹²⁻¹⁵. Direct electrophysiological recording in animal models³⁶⁻³⁹ and fMRI in humans¹⁰ have observed neural activity in LC in response to unexpected and abrupt contextual changes. Pharmaceutical evidence has demonstrated that downregulating NE results in impaired adaptation to environmental changes^{4,40} (but see^{41,42}) whereas pharmacologically stimulating the noradrenergic system is associated with increased learning rates^{43,44}. Indirect measures of NE release, based on pupillometry, have also revealed an association between NE signaling and increased learning rates – a proxy for model resetting^{2,11}. However, the existing literature is limited by the fact that most of the experimental results taken to support the 'NE as an interrupt signal' hypothesis have

involved tasks in which inputs evolve slowly over time and participants make either active decisions about stimulus predictability or are required to form stimulus–response associations^{2,4,10,11,20,21,28}. In contrast, here we used rapid (20Hz), not explicitly trackable, auditory patterns, specifically structured to evoke model resetting vs. model updating. We demonstrate that when pattern changes were behaviourally irrelevant, pupil dilation responses (PDR) were evoked exclusively by changes associated with violations of regularity, thus establishing that NE plays a role in coding model violations on time scales relevant to tracking unfolding sensory information, even when it is not behaviorally relevant. We further demonstrate that behavioural relevance exerted a major effect on pupil dynamics, changing the responses both during the establishment of patterns, and at transition periods. Overall, the results are consistent with a hypothesized role of NE as a model interrupt signal, and provide a rich view of the contingencies that have automatic and/or controlled access to this interrupt.”

Critically, we use the present stimuli as a tool with which to study whether:

- (1) The role of the pupil-linked LC-NE system in monitoring the statistics of the environment extends to rapid sensory signals**
- (2) Whether/how these processes are affected by behavioural relevance.**

As detailed in the introduction (and recapitulated in the discussion section), the RAND-REG / REG-RAND stimuli are useful for this purposes because they provide an elegant method of dissociating transitions associated with model resetting from those associated with model updating, whilst maintaining identical perceptual detectability. We now state this explicitly in the beginning of the results section (p 6):

“Brain response (MEG and EEG) data suggest that while REG₁₀-RAND₂₀ and RAND₂₀-REG₁₀ transitions are characterized by opposite statistics (emergence vs. violation of regularity; see also Figure S1 for an information theoretic characterization of the sequences), both are detected automatically, and at a similar latency even when participants’ attention is directed elsewhere^{22,24}. When asked to respond behaviorally to transitions, listeners exhibit ceiling performance and similar reaction times with comparable variability²² (also replicated here in Exp. 3). Thus, these signals are well suited for dissociating transitions associated with model resetting from those associated with model updating and provide an elegant method for disambiguating the role of the pupil linked LC-NE system in tracking statistics of rapidly-evolving sensory signals. “

Our results indeed demonstrate that the asymmetry we observed, and which we interpret in terms of NE specificity, is not related to detectability but rather to the statistics of the stimulus. The prediction is that any sequence structure (including at very rapid rates) that is associated with abrupt model violation should evoke a PDR. Any sequence structure that is associated with a smooth change in statistics should not.

The model we use is a simple model of sequence tracking. While it matched human behaviour on some respects (see also Barascud et al 2016), it failed to capture others. For example, the model treats RAND-REG₁, RAND-REG₂, RAND-REG₅, RAND-REG₁₀ similarly whereas the pupil responses to these transitions are clearly different. This suggests that pupil response data can be used towards understanding the manner in which signal statistics are represented in the brain. We hope that our results will motivate future work in this direction.

Q2: I understand that the authors have seen 4 null results regarding baseline effects, but as I said in my previous review, null effects should not be interpreted as evidence for the null, but failure to reject it. One reason that the null might not be rejected is due to lack of power. One reason that the baseline analyses provided in the current manuscript may lack power is that they include low frequency variability in the pupil response that is unrelated to the underlying biological processes of interest (eg. Reimer 2015). I don't understand the authors' response to my concern regarding baseline subtraction. There is no reason that the regression approach cannot be taken on categorical data. I am not saying that the authors need to change analyses throughout the paper, but given that baseline subtracted results can pick up on either side of the equation, it would be nice to clearly demonstrate that the effects that they are interested in emerge only from the positive side of the subtraction. As I said before, this could be done with a regression, or it could be done by showing that the category differences hold in PDR after regressing out pre-transition baseline from their PDR measure. Based on the data in R4, it seems unlikely that removing variance related to pre-transition baseline in the PDR will substantially change the results, but given that doing so is critical for interpretation, and that figure R4 is not included in the paper, it seems critical that the authors make the appropriate statistical argument to show that their effects are not driven by baseline differences.

A2. We apologize for misunderstanding the original suggestion. We have now performed the analysis requested by this reviewer (as well as reviewer 3, below). As expected, whilst some values (e.g. precise timing of effects) changed slightly, the qualitative results remain the same as in the previous version. We now report the new analysis results in the paper, instead of those which used traditional baseline correction.

Specifically, the relevant part in the methods section now reads (p 52):

“To remove variance related to the pre-transition baseline, baseline pupil size (mean value over the baseline interval) was regressed out in a point-by-point manner from the data for each stimulus condition in each subject (regression coefficients were computed independently for each sample point in each condition). Thereafter, the obtained time series (residuals after accounting for linear baseline dependences) were time-domain-averaged across all epochs of each condition type to produce a single time series for each condition. Matched no-transition conditions were processed in a similar manner around ‘dummy’ transition times set to match those in the transition conditions. In a separate analysis (not shown) we also employed the standard baseline correction approach (direct subtraction of the mean baseline pupil size; i.e. assuming that the regression coefficient =1). This yielded qualitatively identical results”.

The figures and results (numerical values for latencies etc) have been updated throughout to reflect the new analysis. We also removed Figure S3 (Single-trial relationship between pre transition amplitude and PDR) as it is no longer needed. But are happy to put it back if the reviewers consider it necessary.

Reviewer #3

Q1: The authors may have misunderstood the comments reviewer 2 and I raised regarding baseline pupil. We highlight that PDRs are inherently dependent on the baseline size of the pupil on each trial. For example, PDRs will be smaller if baseline pupil is larger, simply by

virtue of the fact that the PDR has less room to increase when the pupil is already dilated. To account for the influence of baseline pupil on the PDRs, the authors should include baseline pupil (the actual baseline value) as a nuisance regressor in their PDR pupil analysis (similar to Krishnamurthy et al., 2017, Nat. Hum. Behavior) – i.e., in predicting the pupil change from baseline at each discrete timepoint in their PDR analyses, the authors could include a nuisance regressor of the value of the baseline pupil diameter.

A1: Apologies for the misunderstanding. The value of the baseline pupil diameter has now been included as a regressor in the analysis. Please see our response to a similar comment by reviewer 2, Above (Q2).

Q2: I still think more should be discussed regarding whether the effects the authors observe, particularly the difference between Exp. 1 & 2 and Exp 3, can be broadly characterized as unexpected uncertainty. As I've stated in my previous comments, Exp 1 & 2 do not require some internal model of the stimuli, suggesting to me that the effects the authors observe are linked more to sensory effects (e.g., some form of repetition suppression) rather than violations of top-down expectations. When top-down expectations become a part of the task (e.g., Exp 3), the asymmetry between RAND-REG and REG-RAND disappears, providing stronger evidence for the role of unexpected uncertainty. I do find this distinction interesting and important, and feel like it should get more attention in the authors interpretations of their results.

A2: We do not think that the PDR asymmetry observed in the current study can be explained by pure sensory effects. For example, in Exp2, we report PDRs evoked by REG-RAND transitions which constitute pure pattern violations (i.e. where the transition is manifested as a change in pattern without the introduction of frequency transients). For such transitions to be detected, the brain must quickly (within the 2-3 seconds which precede the transition) acquire a representation of the unfolding regular pattern, monitor the ongoing sequence, and detect when it has been violated. It is difficult to account for this ability (especially for complex patterns such as REG10) as a form of repetition suppression.

Our previous brain imaging work with similar stimuli (Barascud et al, 2016; Southwell & Chait, 2018) under passive listening (i.e no behavioural relevance) conditions has revealed that regularity extraction and detection of pattern violations is sub-served by a network of areas, including frontal loci (IFG and orbito-frontal cortex), consistent with the involvement of these areas in model maintenance and suggesting that violations of patterns involve violation of top-down expectations. We discuss this literature in the paper (“relationship to brain responses” section in the discussion).

Importantly, these experiments also demonstrate robust brain responses to RAND-REG transitions (see also reproduced in Figure 1) suggesting that the PDR asymmetry observed here is not due to the fact that RAND-REG transitions are not being detected. We have added a statement about this in the discussion section (p 39) which now reads:

“Importantly, the presence of extensive brain activation to behaviorally irrelevant RAND-REG transitions, suggests that the lack of a PDR in that condition is not due to those transitions not being detected in the passively listening brain. Rather, it appears that this information is not conveyed to the LC-NE system.”

With respect to the effect of behaviour on the PDR: Whilst, we cannot make conclusive statements about the source of the distinction between Expts 1 & 2 and Expt 3, we discuss several alternatives in the text (page 39):

“For example, it is possible that task relevance, or heightened arousal under behaviourally relevant conditions, leads to a richer representation of the statistics of the RAND patterns or contributes to the emergence of a category boundary between REG and RAND patterns thereby rendering the transitions, in both directions, as model violations this evoking “unexpected uncertainty” for RAND-REG. Alternatively, behavioral relevance may alter the boundary between “expected” and “unexpected” uncertainty, resulting in a threshold change for model reset”.

We have also re-organized the discussion section which focuses on the specificity of the PDR under passive listening conditions. We hope that this makes our interpretation of the results of Experiment 4 (RAND-REGx) more explicit. The relevant text now reads: (page 36)

“To gain further insight into the statistical model building process for $RAND_{20}$, we systematically investigated $RAND_{20}$ -REG transitions by modulating REG cycle length (Exp 4). A default hypothesis was that since all have similar statistical structure (are manifested as a smooth change in information content; see figures 8, S2), they should all result in effects identical to that observed for $RAND_{20}$ -REG₁₀ - namely not evoke a PDR. We indeed observed a reduced PDR for $RAND_{20}$ -REG₂ onwards which we interpret as further evidence for the specificity of the pupillary response.

Interestingly, the $RAND_{20}$ -REG₁ transition did evoke a PDR. This may be taken to indicate that, unlike for REG with longer cycles, a transition to REG₁ is treated as an abrupt model violation with respect to the internal model maintained for $RAND_{20}$. One suggestion is that the brain may engage in a form of automatic latent model building using just the last few tones. If the latent model based on those few tones fits them much better than the prevailing model, then an abrupt change is reported. Under this hypothesis, the fact that even as simple a sequence as two alternating tones does not generally lead to model change detection suggests stringent constraints on the automatic model construction – perhaps that it encompasses no more than two successive tones. It is tempting to speculate that such model construction could be implemented by low-level coding mechanisms e.g. adaptation or repetition suppression, both of which would lead to detectably unusual patterns of activity in tonotopically organized neural populations.

Overall, the results indicate that the presence of a PDR, as a marker for “unexpected uncertainty”, can be used to probe the models observers automatically construct of their surroundings. The demonstration of a distinct boundary between ‘expected’ and ‘unexpected’ uncertainty under behaviorally-irrelevant listening conditions calls for future modeling and experimental work to outline the properties of this distinction and its implications for perception.”

Q3: Regarding the p values for the linear mixed effects models, the authors indicate in the methods that these models were run in R using the lmer package, which does not, by default, provide a p-value. How were the p-values computed? I ask because p-values can be computed using likelihood ratios, but these have a tendency to increase type 1 error. The preferred method for computing p-values for LMEs is by using approximations such as the Satterthwaite method for computing degrees of freedom (see Kuznetsova, et al., 2018, J. Stat. Soft. – lmerTest package).

A3: Thank you for the specific query - as we now clarify in Methods, the linear mixed effects models were fit using REML in JMP 13.2 (SAS Institute) whereas the partial R² values were estimated in R using r2glmm. JMP also uses the Satterthwaite

approximation for this family of models (SAS Institute Inc. 2017, ch. 79). We also verified that the results of model tests using JMP converged with those using lme4 and lmerTest - findings were qualitatively the same, with only minor differences in df and p-value estimation. We now include this additional information in methods.

Reviewers' Comments:

Reviewer #2:

Remarks to the Author:

The authors have addressed my concerns.

Reviewer #3:

Remarks to the Author:

The authors have done a good job of responding to the remaining concerns. It might be helpful to add some more details about how they account for baseline effects on pupil diameter, because even though it seems clear and reasonable in their response, it still is not obvious to me from the manuscript what they did. For example, in Methods they now state: "To remove variance related to the pre-transition baseline, baseline pupil size (mean value over the baseline interval) was regressed out in a point-by-point manner from the data for each stimulus condition in each subject (regression coefficients were computed independently for each sample point in each condition)." What exactly is the baseline interval that was used? In the figures showing pupil, are the data labeled "Pupil diameter change from baseline [z-score]" really the residuals after this baseline correction? Why don't they start at zero (presuming the baseline epoch just precedes that time)?

Response to reviewers - NCOMMS-18-35543B

Please find below our response to the final comment from Rev3. We are grateful for everybody's time in commenting on this work, and appreciate the resulting improvements to the final manuscript.

REVIEWERS' COMMENTS:

Reviewer #3 (Remarks to the Author):

The authors have done a good job of responding to the remaining concerns. It might be helpful to add some more details about how they account for baseline effects on pupil diameter, because even though it seems clear and reasonable in their response, it still is not obvious to me from the manuscript what they did. For example, in Methods they now state: "To remove variance related to the pre-transition baseline, baseline pupil size (mean value over the baseline interval) was regressed out in a point-by-point manner from the data for each stimulus condition in each subject (regression coefficients were computed independently for each sample point in each condition)." What exactly is the baseline interval that was used? In the figures showing pupil, are the data labeled "Pupil diameter change from baseline [z-score]" really the residuals after this baseline correction? Why don't they start at zero (presuming the baseline epoch just precedes that time)?

We believe the confusion arose from our reference to z scoring in the y axis labels. Z scoring is a standard pre-processing step, applied to all trials within an experimental block for the purpose of facilitating comparison across subjects. We have now removed this from the y axis labels but expanded the relevant point in the methods (page 24) which now reads:

"To allow for comparison across trials, subjects and experiments, data for each subject in each block were z-normalized based on the mean and standard deviation computed across all the data (all epochs, all conditions) within the block."

We have also clarified the baseline interval (page 24) which now reads:

"To remove variance related to the pre-transition baseline, baseline pupil size (mean value over the 1-second-pre-transition interval) was regressed out in a point-by-point manner from the data for each stimulus condition in each subject (regression coefficients were computed independently for each sample point in each condition)."

Plotted are indeed the residuals after the baseline correction as was already stated in the methods section (page 24):

"Thereafter, the obtained time series (residuals after accounting for linear baseline dependences) were time-domain-averaged across all epochs of each condition type to produce a single time series for each condition."

Note that since we are regressing the mean baseline pupil size (i.e., a single value) the baseline period is not expected to be exactly at 0 for the full duration of the baseline, but instead fluctuate around 0 which is what we demonstrate in the figures. We plot the baseline period for transparency, and it is clearly stated in all figure legends that 'x=0' marks the onset of the sequence/transition as appropriate.